# Atmospheric River Representation in the Energy Exascale Earth System Model (E3SM) Version 1.0

Sol Kim[1,2], L. Ruby Leung[2], Bin Guan[3,4], and John C. H. Chiang[1]

[1]Department of Geography, University of California, Berkeley, CA, USA
[2]Pacific Northwest National Laboratory, Richland, WA, USA
[3]Joint Institute for Regional Earth System Science and Engineering, University of California, Los Angeles, CA, USA
[4]Jet Propulsion Laboratory, California Institute of Technology, Pasadena, CA, USA

**Correspondence:** Sol Kim (solkim@berkeley.edu)

**Abstract.** The Energy Exascale Earth System Model (E3SM) Project is an ongoing, state-of-the-science Earth system modeling, simulation, and prediction project developed by the U.S. Department of Energy (DOE). With an emphasis on supporting DOE's energy mission, understanding and quantifying how well the model simulates water cycle processes is of particular importance. Here, we evaluate E3SM version v1.0 for its ability to represent atmospheric rivers (ARs), which play significant roles in water vapor transport and precipitation. The characteristics and precipitation associated with global ARs in E3SM at standard resolution (1° x 1°) are compared to the Modern-Era Retrospective analysis for Research and Applications, Version 2 (MERRA2). Global patterns of AR frequencies in E3SM show high degrees of correlation ($>=0.97$) with MERRA2 and low mean absolute errors ($<1$ %) annually, seasonally, and across different ensemble members. However, some large-scale condition biases exist leading to AR biases - most significant of which are: the double-ITCZ, a stronger and/or equatorward shifted subtropical jet during boreal and austral winter, and enhanced northern hemisphere westerlies during summer. By comparing atmosphere-only and fully-coupled simulations, we attribute the sources of the biases to the atmospheric component or to a coupling response. Using relationships revealed in Dong et al. (2021), we provide evidence showing the stronger north Pacific jet in winter and enhanced northern hemisphere westerlies during summer associated with E3SM's double-ITCZ and related weaker AMOC, respectively, are significant sources of the AR biases found in the coupled simulations.

## 1 Introduction

Atmospheric rivers (ARs) are central actors in the global water cycle and have significant human impacts. These features are narrow, filamentary structures of concentrated water vapor transport in the lower atmosphere responsible for transporting the majority of water vapor across the mid-latitudes towards the poles (Zhu and Newell (1998)). Recently, ARs have received a categorization similar to hurricanes which describes the wide range of possible AR impacts - both beneficial and destructive (Ralph et al. (2019)). Weaker ARs, with integrated vapor transport (IVT) values of around 250 kg m[-1] s[-1], can provide regions such as the west coast of the U.S. with critical sources of precipitation, while exceptional ARs, with IVT values well over 1250 kg m[-1] s[-1], can be associated with widespread flooding and hazards to both human life and infrastructure. A recent study examining the last 40 years of floods in the western U.S. found ARs pose a $1 billion-a-year flood risk (Corringham et al.

(2019)). Many studies indicate that ARs will increase in frequency and/or intensity and will deliver more precipitation under global warming (e.g. Payne and Magnusdottir (2015); Espinoza et al. (2018); Payne et al. (2020); O'Brien et al. (2021)). On the west coast of the U.S. for example, ARs are expected to increase the occurrence of extreme precipitation and associated flood risk, including via their contribution to snow/ice melt (Swain et al. (2018); Chen et al. (2019)). With such important socioeconomic impacts, increasing our understanding of ARs in past, present, and future climates is critical to mitigate damage and protect life and property.

Although ARs have been directly observed since the late 1990s with aircraft and dropsondes (e.g. the California Land-falling Jets Experiment (CALJET) (Ralph et al. (2005)) and via satellites with passive microwave radiometers (Ralph et al. (2004); Ralph et al. (2006)), there still exists gaps and challenges to direct observations of ARs due to both the scale and the extreme environments associated with ARs. While efforts to directly observe ARs continues to be improved, much of these efforts to date have been regionally specific to the western U.S., where ARs play a critical role in water resources. Thus, many researchers have instead relied on the use of gridded reanalysis products - which incorporate a variety of observations - to study ARs in historical, regional, and global perspectives (Ralph et al. (2020)). Given the socioeconomic impacts of ARs, there is also wide and increasing interest on the behavior of ARs in future climates which typically require the use of global climate models (GCMs) (e.g. Dettinger et al. (2011); Payne and Magnusdottir (2015); Warner et al. (2015); Shields and Kiehl (2016)). A critical step in using these GCMs, which are used to simulate ARs in a variety of climates, is to first establish confidence in the model's ability to simulate ARs in the current climate.

Many efforts have already evaluated a large array of different models. For example, Guan and Waliser (2017) evaluated 22 GCMs that participated in the Global Energy and Water Cycle Experiment (GEWEX) Atmospheric System Study (GASS)-Year of Tropical Convection (YoTC) Multimodel Experiment which included both atmosphere-only and ocean-atmosphere-coupled models for the period of 1991-2010. They used reanalysis products to quantify model errors in the context of reanalysis uncertainty and found large errors across all models. Another study by Espinoza et al. (2018) evaluated 21 GCMs in the Coupled Model Intercomparison Project Phase 5 (CMIP5) for their representation of ARs in historical and two future climates (Representative Concentration Pathway, or RCP, 4.5 and 8.5). The multimodel mean (MMM) was a good representation of their reference dataset, ERA-Interim, but tended to have a general underestimation of AR frequencies in the midlatitudes. Additionally, intermodel differences showed significant disagreement for AR frequencies in the subtropics. Payne and Magnusdottir (2015) performed a similar analysis on landfalling ARs to understand responses to warming in RCP8.5 and compared CMIP5 historical runs to both ERA-Interim and the Modern-Era Retrospective Analysis for Research and Applications (MERRA) as an initial step. Most models were able to resolve the general shape of wintertime landfalling AR frequencies but only a few could resolve other characteristics such as interannual variability in amplitude of moisture flux and median landfalling latitude. A strong relationship between model biases in the North Pacific subtropical jet and landfalling AR frequency on the west coast of North America has been identified based on analysis of large ensemble simulations from Community Earth System Model (CESM) (Hagos et al. (2016)). More recently, O'Brien et al. (2021) evaluated several CMIP5 and CMIP6 models' historical simulations (prior to evaluating a projection scenario) against MERRA2 and found frequency distributions to be remarkably consistent.

The United States Department of Energy (DOE) recently released the Exascale Energy Earth System Model version 1 (E3SMv1) which is a state-of-the-science Earth system model. The model was developed to support the DOE's energy mission, with an emphasis on modeling the long-term changes in air and water temperatures, water availability, storms and heavy precipitation, coastal flooding and sea-level rise on high performance computers (Leung et al. (2020)). The standard resolution model (1° x 1°) has been shown to credibly simulate earth's climate when evaluated by means of a standard set of Coupled Model Intercomparison Project Phase 6 (CMIP6) Diagnosis, Evaluation, and Characterization of Klima (DECK) simulations which include a preindustrial control, historical simulations, and idealized $CO_2$ forcing simulations (Golaz et al. (2019)). A suite of atmospheric fields (e.g. net top-of-the-atmosphere radiation, surface air temperature, zonal winds, and precipitation) in the historical E3SMv1 simulations were compared against observations to calculate root-mean-square-errors (RMSEs). When compared to an ensemble of 45 CMIP Phase 5 (CMIP5) models, E3SMv1's RMSEs were generally found to have lower errors than the median of the CMIP5 ensemble, and for many fields and seasons, in the lowest (best) quantile. There are however, known biases in E3SMv1 which are common to other GCMs, such as a reduction in cloudiness over the subtropical stratocumulus regions and the well-known double Intertropical Convergence Zone (double-ITCZ) issue where there is excessive southern central Pacific precipitation (Zhang et al. (2007); Golaz et al. (2019)).

Evaluating E3SMv1 for ARs, which has not yet been done, is necessary given our need to understand issues surrounding the water cycle and its interactions with humans and other Earth systems. This paper aims to provide an overview of ARs simulated in E3SMv1 by: i) comparing against historical ARs detected in MERRA-2 (Modern Era Retrospective analysis for Research and Applications, version 2) to identify AR biases, ii) evaluating internal model AR variability using individual ensemble members, and iii) determining the large-scale and model sources of AR biases. ARs in this study are detected using the AR algorithm developed by Guan and Waliser (2019) which is a widely used algorithm and has been demonstrated to closely match key AR characteristics from direct airborne observations (Guan et al., 2018). The structure of this paper is as follows. In Section 2, E3SMv1, the reanalysis data set, and the AR detection algorithm are described. AR frequency, characteristics, precipitation, and large-scale conditions in E3SMv1 are compared to MERRA2 in Section 3. Discussion and conclusions are presented in Section 4. Appendix material are contained in Section 5.

## 2    Data and Methods

### 2.1    Exascale Energy Earth System Model

E3SMv1 was developed from the Community Earth System Model (CESM1) (Leung et al. (2020)). This study uses global, daily mean data from the standard 1° x 1° resolution (also referred to as the 'low' resolution), fully-coupled E3SMv1 model (Golaz et al., 2019). We use 35 years (1980-2014) from the historical simulation which incorporates several historical, observed forcings including atmospheric composition changes. Five ensemble members of historical simulations are available from E3SMv1 in the CMIP6 archive; these five members use initial conditions from the preindustrial control run, branched out at 50 year intervals, beginning with January 1st of year 101. The ensemble members are used in this study to examine internal variability related to ARs and to generate ensemble mean frequencies of ARs. The E3SM Atmosphere Model (EAM)

(Rasch et al., 2019), which was developed from the Community Atmosphere Model version 5 (CAM5), uses a spectral element dynamical core and is applied at a horizontal resolution of approximately 110 km (or 1° x 1°; 180 latitude grids x 360 longitude grids) and has 72 vertical levels. The historical runs follow the CMIP6 protocols outlined in Eyring et al. (2016).

To understand the sources of model biases in simulating ARs, we also analyze atmosphere-only simulations for comparison with the fully-coupled historical simulations as has been done in previous studies to identify sources of errors (e.g. Li and Xie (2012); Li and Xie (2014)). The atmosphere-only simulations follow the Atmospheric Model Intercomparison Project (AMIP) protocol and are part of the CMIP6 DECK simulations. All three available AMIP ensemble members are used in this study and atmosphere and land initial conditions for these ensemble members were taken at year 1870 from the first three histor-

ical ensemble members. The AMIP simulations have prescribed SSTs and sea ice concentrations from observations. A full overview of E3SMv1 and EAM can be found in Golaz et al. (2019) and Rasch et al. (2019) respectively. We will henceforth refer to E3SMv1 as E3SM. The fields obtained from E3SM are: total (vertically integrated) zonal/meridional water flux, total (convective and large-scale) precipitation rate (liquid + ice), total (vertically integrated) precipitatable water, zonal wind at 200 hPa, and geopotential height at 500 hPa.

## 2.2  Reanalysis Data Set

In this study, daily mean reanalysis data from MERRA2 (Gelaro et al. (2017)) is analyzed for the same 35 year period as E3SM (1980-2014). The native spatial resolution is ∼50 km (or 0.5° x 0.625°; 361 latitude grids x 576 longitude grids) but was re-gridded to match E3SM to facilitate comparison. MERRA2 is an updated version of MERRA (version 1) which was developed to improve representations of the global water cycle. Good agreement between MERRA2 against airborne

and satellite observations has been previously demonstrated for ARs (Ralph et al. (2012); Guan et al. (2018)). In additon, MERRA2 has been used extensively in previous AR studies (e.g. Shields et al. (2018); Rutz et al. (2019)). The fields obtained from MERRA2 are the same as those obtained from E3SM but can correspond to different long names: eastward/northward flux of atmospheric water vapor, total precipitation, atmosphere water vapor content, eastward wind at 200 hPa, and geopotential height at 500 hPa.

## 2.3  AR Detection Algorithm

ARs are detected using tARget v3 - the latest version of a widely used algorithm developed for global studies. Details of the detection algorithm can be found in Guan and Waliser (2015), Guan et al. (2018), and Guan and Waliser (2019). This algorithm is part of the Atmospheric River Tracking Method Intercomparison Project (ARTMIP) (Shields et al., 2018) and is among the relatively 'permissive' algorithms compared to other algorithms to facilitate global analyses including inland-

penetrating ARs as well as polar ARs, but meanwhile effective in filtering out non-AR features in the tropics. While there is significant uncertainty associated with the choice of detection algorithm (Shields et al. (2018); Shields et al. (2019); O'Brien et al. (2020); Rutz et al. (2019)), we choose to use this algorithm for its global applicability and make no attempt in this study to quantify the uncertainty as this is a primary goal of ARTMIP. The algorithm, when applied to contemporary reanalysis products, detected ARs that were found to closely match airborne observations in terms of key characteristics such as AR width and total

IVT across AR width (Guan et al., 2018). Various refinements and improvements have been made to the algorithm since it
was first introduced in Guan and Waliser (2015) to its current version. As a brief summary, the algorithm extracts contiguous
areas of connected gridpoints based on IVT exceeding location- and season-specific IVT thresholds set at the 85th percentile
of the dataset analyzed but cannot go below 100 kg m$^{-1}$ s$^{-1}$. Geometric and directional requirements are then applied to the
identified objects, with considerations on: direction of object-mean IVT (poleward component >50 kg m$^{-1}$ s$^{-1}$), coherence of
IVT directions (more than half of the area having an IVT direction within 45° from the object-mean IVT), length (>2000
km) and length/width ratio (>2) (Guan and Waliser (2019)). The IVT threshold is calculated separately for MERRA2 and the
E3SM simulations. We include the annual mean 85th percentile IVT of both along with the differences in Section 5 Fig. A2. In
general, E3SM has higher threshold IVT values than MERRA2 with some regional biases up to 100 kg m$^{-1}$ s$^{-1}$. Notably, these
high positive IVT bias regions are over the subtropical jet in the northern hemisphere (which we discuss throughout the paper)
and the midlatitude jet in the southern hemisphere where there are known warm sea surface temperature biases (Golaz et al.
(2019)) and associated enhanced atmospheric moisture (not shown).

## 3   Results

### 3.1   AR Frequency

We begin with the global distribution of AR frequencies shown in Fig. 1. These frequencies represent the ensemble mean from
5 historical simulations for E3SM compared to MERRA2; the AR detection algorithm was applied to each ensemble member
individually to calculate the ensemble mean. For each grid cell, the frequency shown represents the number of timesteps the
grid cell was part of an AR divided by the total number of timesteps in the time period. This is done for the annual, extended
boreal winter (NDJFM), and extended boreal summer (MJJAS). The distribution of the annual frequency of ARs in E3SM
closely matches MERRA2's distribution as seen in Fig. 1a and 1b. In E3SM, as in MERRA2, frequency maxima are found in
the extratropics over the Pacific, Atlantic, and south Indian ocean basins while minima can be seen along the equator, at high
polar latitudes, and over both Greenland and the Tibetan Plateau. The difference of E3SM minus MERRA2 annual frequencies
is shown in Fig. 1c. To note, in the annual (seasonal), one percent of absolute difference translates to 3.65 (∼1.5) timesteps
of AR difference out of 365 (151-153) timesteps in a year (extended season). All percentage differences mentioned below,
unless otherwise noted, are absolute differences, not relative differences. We include the relative differences of regions with
absolute AR frequencies of at least 3 % in Section 5 Fig. A2. Overall in E3SM, there are no to weak biases (<1 % absolute
differences) over the AR frequency maxima regions which translate to correspondingly low relative biases. Positive biases of
higher magnitudes (1-3 %) are seen near the edge of the tropics and subtropics in both northern and southern hemispheres.
E3SM also has positive biases in polar areas, specifically over Alaska, Siberia, and just offshore of Antarctica. These absolute
AR frequency differences translate to relative differences of up to 60 % along the subtropical edge of the Pacific, over India, over
the polar areas near Alaska, and off the Antarctic coast (Fig. A2c). These relative differences, while large, are over regions with
annual AR frequencies of 5 % or less (with the exception of Alaska) where AR activity is expected to be particularly sensitive
to large-scale conditions and we explore contributing factors to these biases, particularly to the Pacific storm track region, in

Section 3.4. Negative biases (1-2 %) are seen in the southern hemisphere subtropics (southwestern region of Australia and over the south Atlantic east of Brazil). Between E3SM (ensemble mean) and MERRA2, the annual frequency mean absolute error (MAE) is 0.60 % and the correlation is 0.98 showing strong agreement in the distribution and magnitude of global AR frequencies.

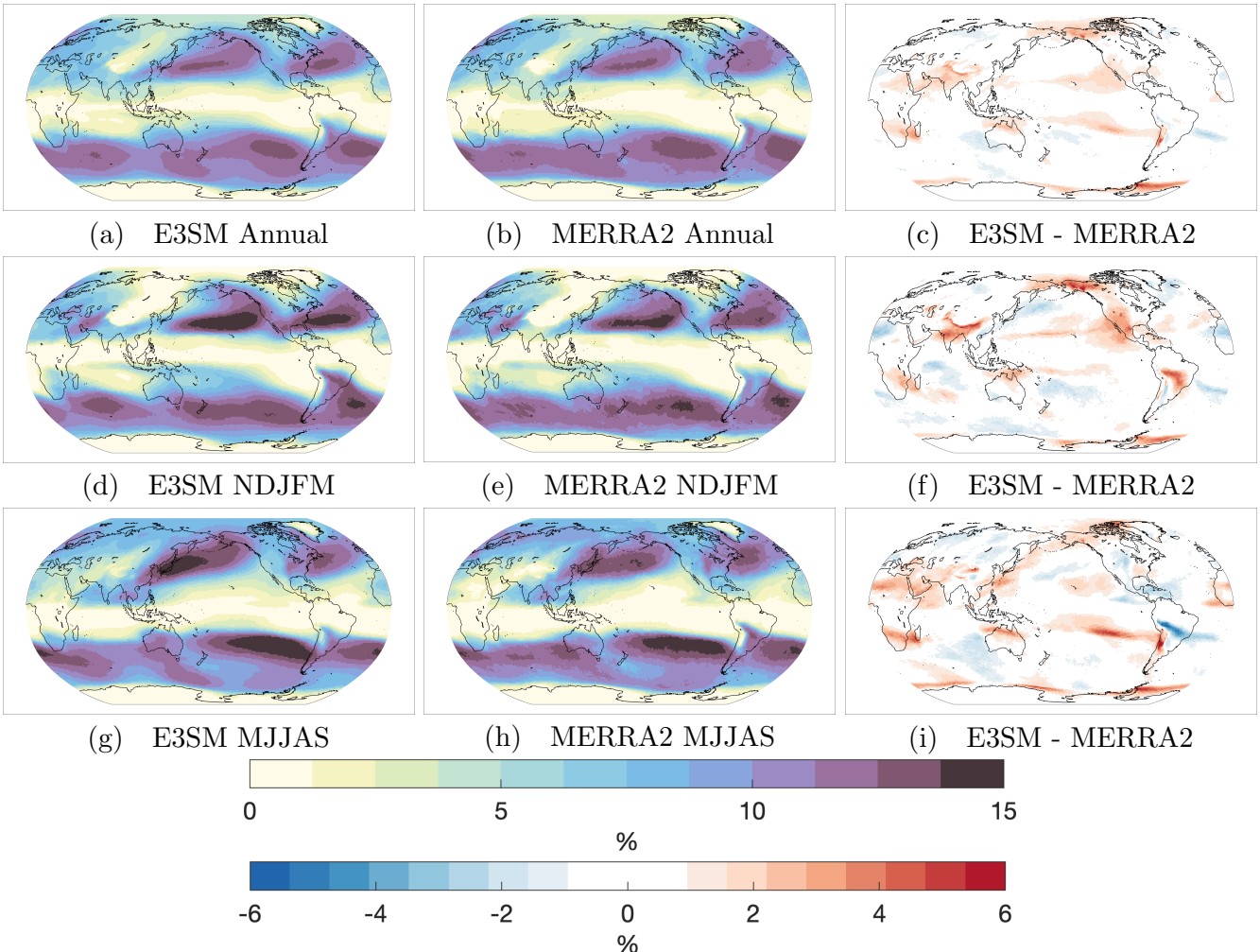

**Figure 1.** The AR frequency at each grid point globally for the annual (top row), extended winter NDJFM (middle row), and extended summer MJJAS (bottom row). E3SMv1 frequencies (left column), MERRA2 frequencies (middle column), and the difference in frequencies between E3SMv1 and MERRA2 (right column). The colorbar on the top (bottom) corresponds to the absolute (difference in) frequencies.

For the NDJFM, E3SM (Fig. 1d) has AR frequency maxima and minima over regions matching MERRA2 (Fig. 1e) but there are areas with seasonal biases (Fig. 1f). ARs are most frequent over the subtropics and midlatitudes over the north and south Pacific and Atlantic where the storm tracks are located. One of the biggest sources of positive biases comes from north Pacific

ARs affecting the entire west coast of North America with frequencies around 3-4 % higher than in MERRA2. The relative differences in this region can reach up to 30-50 % (Fig. A2a) and lead us to investigate biases of the north Pacific subtropical jet in E3SM compared to MERRA2 in Section 3.4. These positive biases stretch from Mexico to Alaska nearly uninterrupted as well as nearly all the way across the subtropical north Pacific basin. Other positive biases exist over South America (originating from the Amazon Rainforest) and over India near the Himalayas with high relative differences (>50 %). Negative anomalies of $\sim$1-2 % are seen throughout the southern hemisphere ocean basins as well as over northern Africa and east of Japan. The NDJFM frequency MAE is 0.72 % and the correlation is 0.98.

The MJJAS frequencies in E3SM (Fig. 1g), similar to NDJFM frequencies, have maxima and minima co-located with MERRA2's MJJAS frequencies (Fig. 1h) but, again, with some biases (Fig. 1i). Frequency maxima can be seen in both E3SM and MERRA2 over the western areas of the north Pacific and north Atlantic basins and over the south Pacific and south Atlantic in the subtropics and midlatitudes. Positive biases are seen for E3SM in the following areas: the southern edge of the tropics near 15° S with the exception of the negative frequency bias over South America, the Middle East, the western boundary of the north Pacific, windward side of the Andes, and offshore of the Antarctic. The regions where these absolute differences translate to large relative differences are around 15° S (exceeding 100 % at the tropical edges), over the Arabian Sea (25-60 %), and off the shore of Antarctica (exceeding 100 % closer to shore). Negative biases are overall weaker and are found over South America on the leeward side of the Andes (between $\sim$10-20° S), the Caribbean Sea, south Indian Ocean, and at mid and polar latitudes over Eurasia. The MJJAS frequency MAE is 0.82 % and the correlation is 0.97.

To summarize, E3SM and MERRA2 AR frequencies show very high correlation (>=0.97) and low mean absolute errors (<1 %) annually and seasonally. The magnitude and distribution of annual and seasonal AR frequencies are consistent with previous studies examining ARs in reanalyses (e.g. Guan and Waliser (2015); DeFlorio et al. (2019)). However, in E3SM, the hemisphere experiencing winter (especially the northern hemisphere) tends to produce positive frequency biases, and corresponding large relative differences, throughout the border of the tropics and subtropics ($\sim$25° N and $\sim$15° S). The northern hemisphere summer features notable positive anomalies throughout the tropics/subtropics over the Middle East and along the western boundary of the north Pacific. Additionally, AR frequencies are higher near some elevated topography (such as the Himalayas, the Alaska range, and Antarctica).

While the previous results are based on the 5-member ensemble mean, we next determine how well the individual historical E3SM ensemble members are able to match MERRA2's AR frequencies using Taylor diagrams (Fig. 2). Taylor diagrams provide a graphical summary of similarity between two patterns using pattern correlation, centered root-mean-square difference (RMSD), and standard deviation (SD). The Taylor diagrams confirm E3SM's ability to accurately simulate present day AR frequencies globally, across ensemble members, in the annual and seasonal with a high degree of similarity to MERRA2. Correlations are above 0.95 for all ensemble members and for the three periods analyzed (NDJFM, MJJAS, and annual) with the annual having the highest correlation. E3SM SDs are consistent with MERRA2's SD; all ensemble members, for all periods, are within 0.18 % of MERRA2. RMSDs in the annual are under 1.0 % while in the seasonal periods, are under 1.5 %. The diagrams also suggest that the internal variability of E3SM historical simulations is small - particularly in the annual - as the 5 ensemble members are tightly clustered.

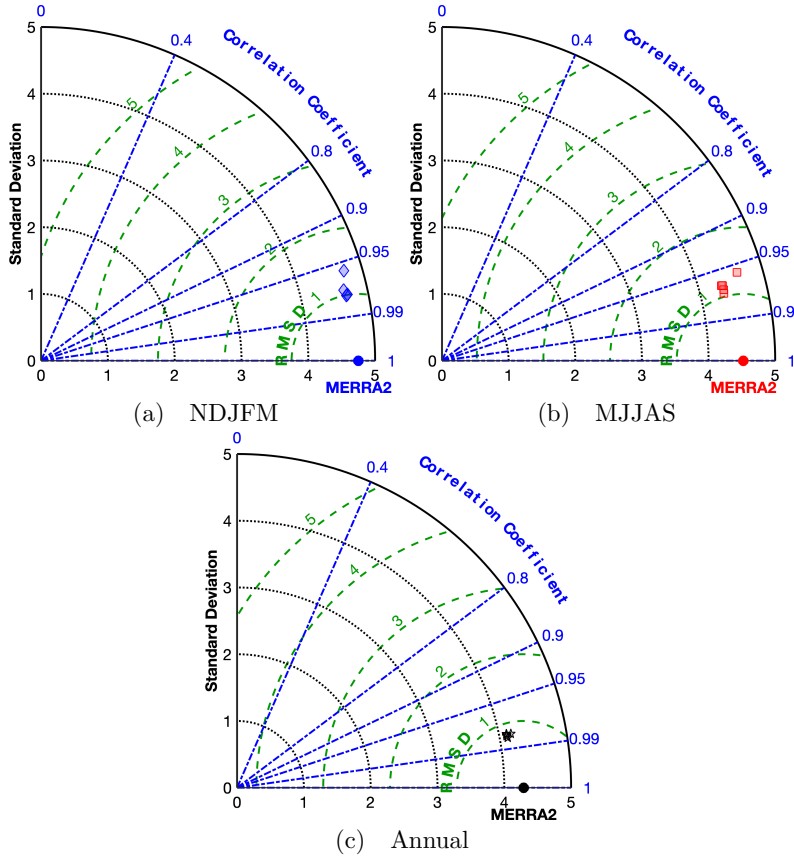

**Figure 2.** Taylor diagrams of AR frequency for the 5-ensemble historical E3SM members against MERRA2 for (a) NDJFM, (b) MJJAS, and (c) annual. The MERRA2 point is labeled in each graph.

As a measure of ensemble spread, the 5-member ensemble SD (different than the Taylor Diagram SD) of AR frequency - i.e. how much variance there is between the individual simulations and the ensemble mean - is shown in Fig. 3. For the annual (Fig. 3c), much of the global SDs are below 0.5 %. A few regions, such as over east and southeast Asia, the Arabian Sea, and the Hudson Bay, have SDs that fall between 0.5 % and 1.0 %. Analysis of the coefficient of variation (not shown), calculated at each grid globally as the ratio between ensemble AR frequency SD and mean ensemble frequency, shows that the annual SDs are well below 10 % of the ensemble mean AR frequencies for virtually all grid points barring a few grid points over the equator and Antarctica where ARs are very rare (annual frequencies are <1.0 %).

During NDJFM (Fig. 3a), SDs are generally higher in the northern hemisphere and have notable peaks of ∼1.5 % over east Asia (which is a jet entrance region) and the north Pacific storm track region. These regional peaks suggest that differences in subtropical jet behavior during NDJFM between the 5 historical simulations may be responsible for some of the internal AR frequency variability. MJJAS SDs (Fig. 3b) have maxima of ∼1.5 % over various regions of the Asia summer monsoon - the Arabian Sea, over the Philippines, and east Asia. This suggests that during MJJAS, differences in monsoon,

MJO, or subtropical jet behavior may drive AR frequency variability between various E3SM ensemble members. In general, the northern hemisphere shows more internal variability; likely due to both internal interannual and interdecadal variability of the midlatitude circulation in the region and especially in the north Pacific.

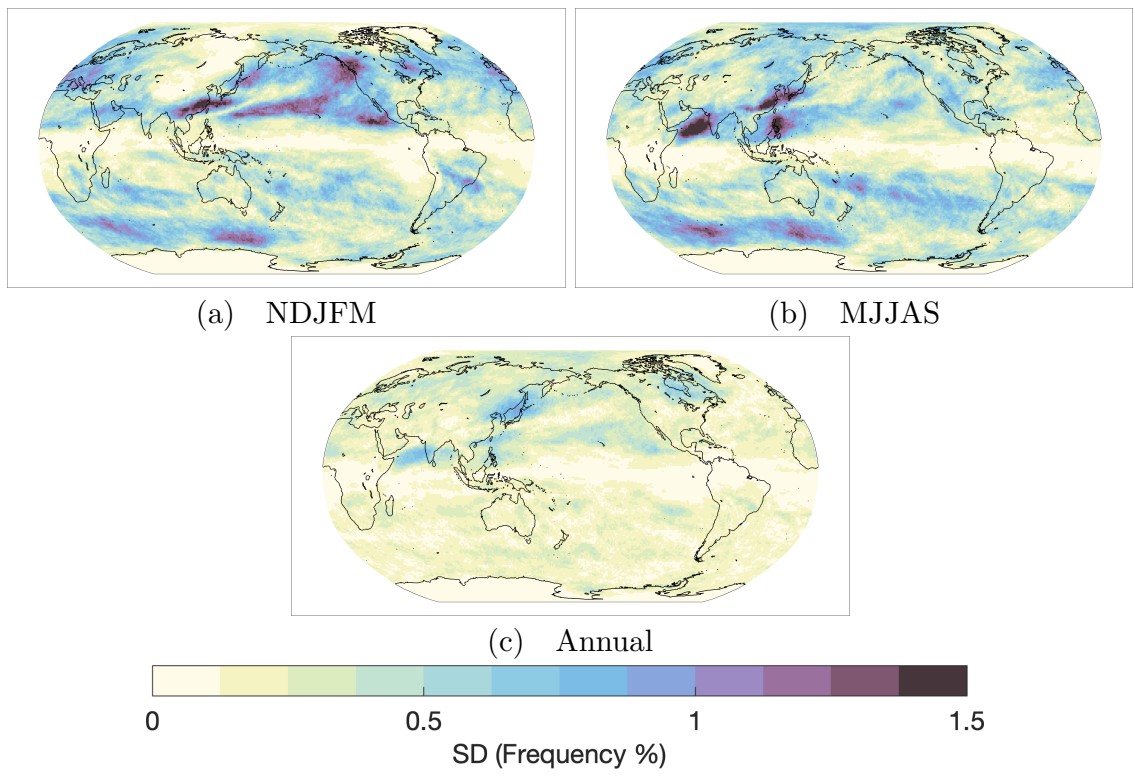

(a)   NDJFM                                    (b)   MJJAS

(c)   Annual

SD (Frequency %)

**Figure 3.** The historical E3SM 5-member ensemble standard deviation of AR frequencies for (a) winter, (b) summer, and (c) annual.

## 3.2   AR Characteristics

Next, we examine several AR characteristics in E3SM (using a single historical simulation) and MERRA2. These characteristics are provided by the AR detection algorithm's output for each individual AR. The distributions for the various AR characteristics are shown in Fig. 4. All characteristics show strong similarities in the distribution shape and peak in probability at the same bin values aside from magnitude of mean IVT (Fig. 4e).

The length (Fig. 4a) and width (Fig. 4b) distributions of ARs in E3SM are generally consistent with MERRA2 as well as with previous characterizations of AR geometry (e.g. Guan and Waliser (2015); Guan et al. (2018)). The median length and width of ARs in E3SM are ∼3 % longer (3501 km compared to 3397 km) and wider (658 km compared to 639 km) than in MERRA2. The E3SM AR length/width ratio distribution and median (Fig. 4c) are in good agreement with MERRA2.

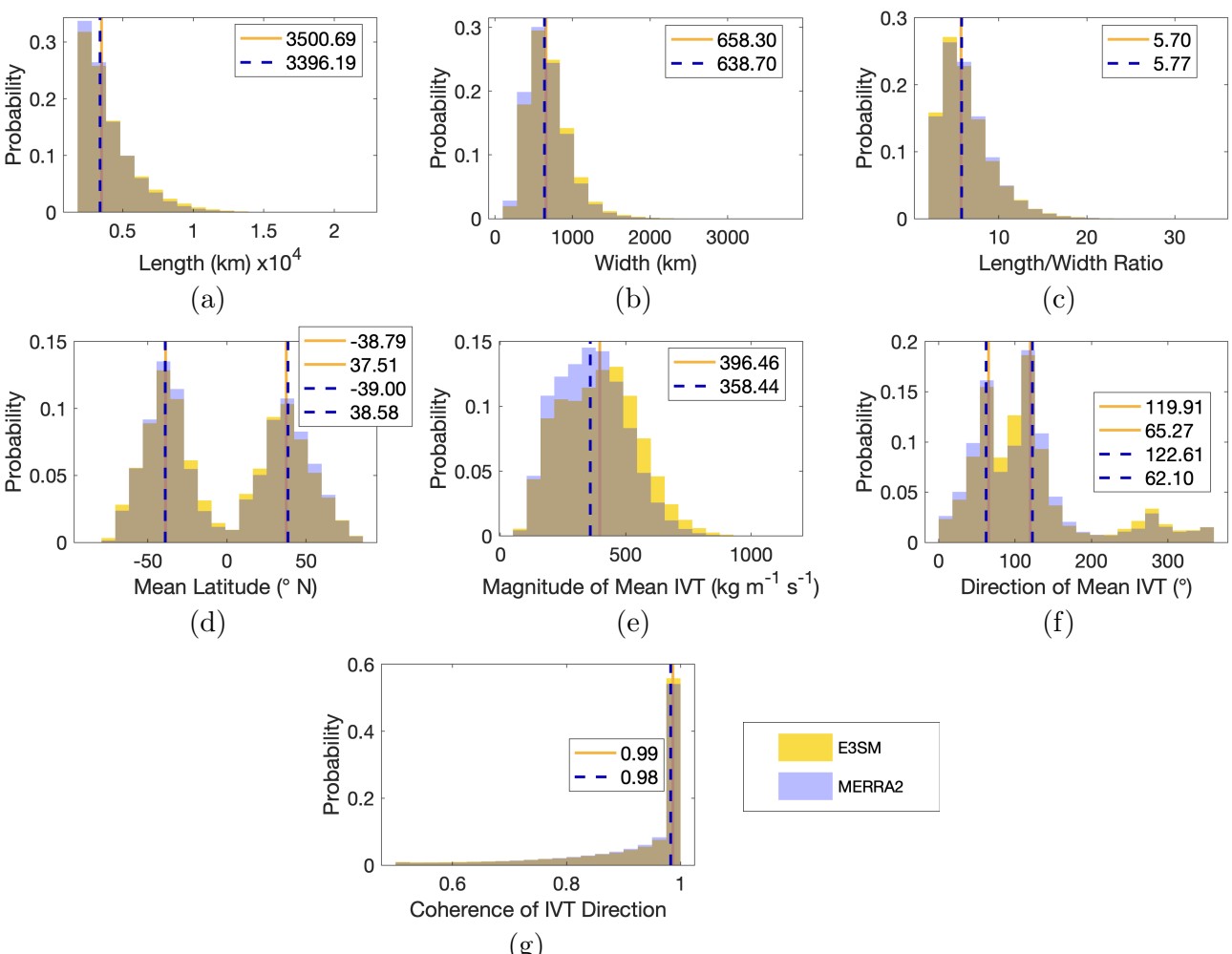

**Figure 4.** Distributions of a variety of AR characteristics in E3SM and MERRA2. The yellow bars and solid line (median) are for E3SM and the blue bars and dotted line (median) are for MERRA2. Two sets of lines indicate hemispheric median values.

The hemispheric median centroid latitude of ARs is consistent, but the distributions reveal that E3SM produces more ARs with centroid latitudes in the tropics, subtropics, and southern polar latitudes while producing fewer ARs in the midlatitudes compared to MERRA2 (Fig. 4d). These results are supported by the frequency differences in Fig.1.

The median magnitude of mean IVT of ARs in E3SM is 10.6 % larger than MERRA2 (Fig. 4e). The distribution of ARs peaks at 500 kg m$^{-1}$ s$^{-1}$ for E3SM but peaks at weaker magnitudes around 350 kg m$^{-1}$ s$^{-1}$ for MERRA2. These distribution differences reflect E3SM's positive IVT biases (Fig. A1c) in regions of high AR activity.

The direction of mean IVT (0° for IVT directed to the north) in the northern hemisphere is directed towards the northeast (median angle of 65° and 62° for E3SM and MERRA2 respectively) and in the southern hemisphere, is directed towards the

southeast (median angle of 120° and 123° for E3SM and MERRA2 respectively) (Fig. 4f). E3SM ARs have higher probabilities around 90° (indicating a mean IVT direction directed to the east) and around 270° (indicated a mean IVT direction directed to the west). These median values, along with the distributions, indicate that E3SM ARs tend to have mean IVTs directed slightly more zonally compared to MERRA2. Coherence of IVT directions within an AR is calculated as the fraction of AR grid cells with IVT directed within 45° of the mean AR IVT (Guan and Waliser (2015)). Model and reanalysis show similarly high coherences of 0.99 (E3SM) and 0.98 (MERRA2) (Fig. 4g).

### 3.3 AR Precipitation

We now compare a variety of metrics related to AR precipitation. For reference, annual precipitation - not just from ARs - is included (Fig. 5a-c) for both models and reanalysis along with the differences. In this study, AR precipitation is defined as the precipitation that falls within an AR boundary.

For annual AR precipitation rates (Fig. 5d-f), E3SM is able to simulate the general global distribution and magnitude of AR precipitation characterized in previous studies using Global Precipitation Climatology Project version 1.2, (e.g. Guan and Waliser (2015); Ralph et al. (2020)) as well as with MERRA2, but with some differences. E3SM has higher AR precipitation estimates over significant portions of the tropics and subtropics but lower estimates along the equator (Fig. 5f) largely reflecting the pattern of general precipitation biases (Fig. 5c). The western coasts of North and South America also have higher rates of AR precipitation in E3SM. The seasonal differences in AR precipitation for the NDJFM and MJJAS are largely consistent to the annual differences. We also include mean annual total precipitation (mm/yr) for E3SM, MERRA2, and their difference (Fig. A3) to show how AR precipitation rate biases translate into yearly quantities. AR precipitation rate biases are important contributors to precipitation totals for regions outside of the tropics where ARs occur more frequently.

Next, we examine the percentage of precipitation attributed to ARs in both datasets annually (Fig. 6). ARs can be responsible for over 30 % of the annual precipitation in the expected extratropical areas such as the west coast of North and South America. Some other areas of note with high AR precipitation fractions include east Asia, the Middle East, southeastern U.S., Greenland, and Australia. These areas of high AR precipitation fractions have been characterized in previous studies (Guan and Waliser (2015); Ralph et al. (2020)). The precipitation fraction differences, shown in Fig. 6c, reveal that E3SM does, however, attribute a higher fraction of precipitation to ARs (up to 20 %) off the coast of southwestern U.S./Mexico and Chile. There is also a strong band of higher AR fractions (exceeding 20 %) extending from the Sahel/Sahara region of Africa eastward to India. In addition, the polar regions exhibit higher AR fractions. Some midlatitude regions attribute less precipitation to ARs in E3SM, particularly over the southern hemisphere oceans. As expected, areas of higher AR precipitation fraction tend to be co-located with areas of positive AR frequency biases. Exceptions are northeast Africa, the Arctic, and over the Amazon - all regions with low annual AR frequency biases; this suggests a small number of AR events are shifting the AR fractions in these locations.

Globally and annually, ARs are responsible for 17.84 % and 17.95 % of precipitation in E3SM and MERRA2 respectively. Interestingly, while the overall precipitation percentage is very consistent, the fraction of AR precipitation that falls over ocean and land vary between the two datasets. In E3SM, 17.38 % (82.62 %) of the AR precipitation falls over land (oceans) while in

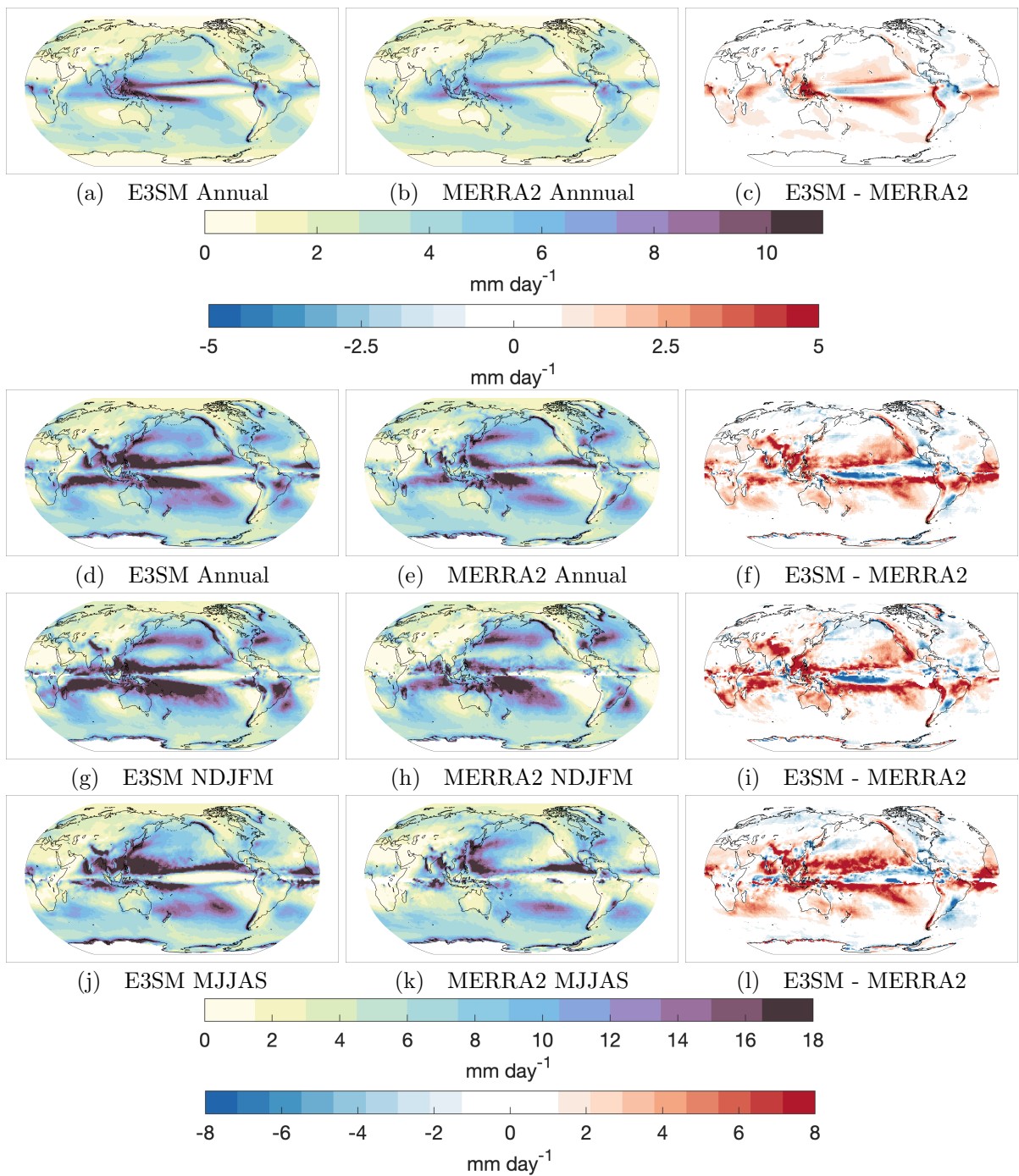

**Figure 5.** Top row shows annual precipitation in E3SM, MERRA2, and the difference. The next three rows are organized as in Fig. 1 but for AR precipitation instead of AR frequency. The colorbar on the top (bottom) corresponds to the absolute (difference in) precipitation or AR precipitation.

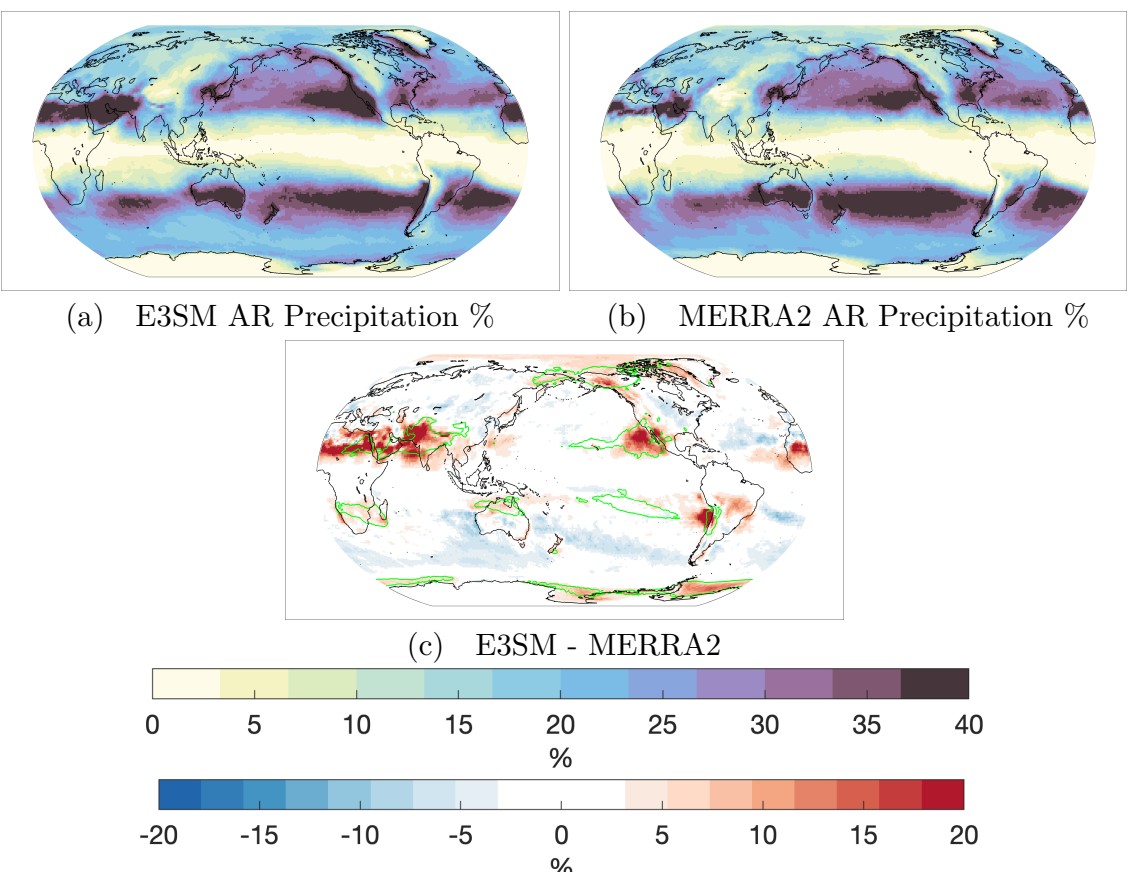

**Figure 6.** The fraction of annual precipitation attributed to ARs for each grid cell for (a) E3SM and (b) MERRA2. The difference (E3SM minus MERRA2) is shown in (c). The colorbar on the top (bottom) corresponds to the absolute (difference) percentages. Contour lines in (c) indicate the 1.5 % positive AR frequency biases from Fig. 1c.

MERRA2, only 14.81 % (85.19 %) falls over land (oceans). Topographic features seem to be able to extract precipitation from AR events more effectively in E3SM compared to MERRA2 - also evidenced by Fig. 5f.

### 3.4 Large-scale AR Conditions

In this section, we look into the sources of the E3SM AR frequency and precipitation biases by examining the large-scale conditions relevant to ARs. We begin with the AR precipitation biases in the fully-coupled E3SM simulations. The E3SM AR precipitation biases (Fig. 5f) are mostly well co-located and of similar magnitude with the general precipitation biases present in E3SM (Fig. 5c). While this reflects the important contributions of AR precipitation to the total precipitation in some regions, it also suggests that the AR precipitation and total precipitation biases can share similar sources of large-scale circulation biases. For example, model biases in the subtropical jet could affect precipitation produced by AR and non-AR storms, as both

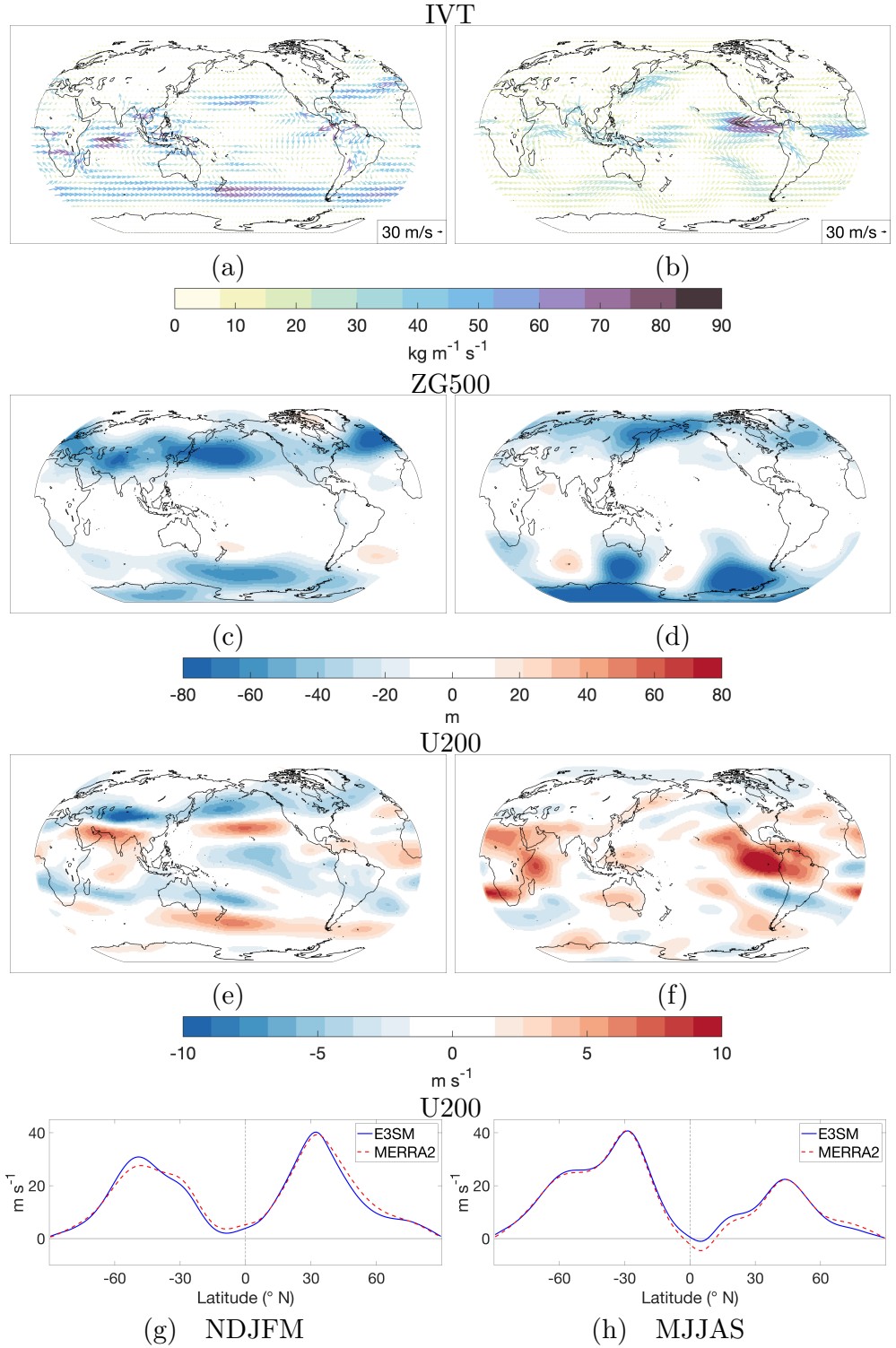

**Figure 7.** Seasonal differences between E3SM and MERRA2 (from top row to bottom row): IVT, geopotential height at 500 hPa, zonal wind at 200 hPa, and zonal means over the Pacific and Atlantic basins (100-360° E) of zonal wind at 200 hPa.

are influenced by the jet and storm tracks (Shields and Kiehl (2016); Zhang and Villarini (2018); Wahl et al. (2019); Ralph et al. (2020)). Two notable exceptions where there are positive AR precipitation biases but no equivalent general precipitation biases are i) off the coast of the U.S. southwest and ii) the region around Pakistan/India. These are the same regions in E3SM, compared to MERRA2, that attribute a higher fraction of the annual precipitation to ARs (Fig. 6), partly due to positive AR frequency biases. This suggests certain large-scale circulation biases in these regions have a larger influence on AR frequency and intensity than the frequency/intensity of non-AR storms leading to a larger fraction of annual precipitation being delivered in the form of AR events rather than non-AR storms.

A large source of general E3SM precipitation biases come from known and common biases in fully-coupled simulations - the double-ITCZ bias and excessive precipitation over the maritime continent (Golaz et al. (2019)). Related to the double-ITCZ, Dong et al. (2021) investigated models with this bias and found that models which feature a present-day double-ITCZ bias tend to exhibit an excessively wet U.S. southwest and understate the drying over the Mediterranean basin in global warming projections. The former on the U.S. southwest wetting is due to these interconnected relationships: under global warming, models with double-ITCZ bias feature enhanced central Pacific rainfall as a wet-get-wetter response, which increases the upper-tropospheric heating in the Pacific subtropics and the meridional temperature gradients, resulting in an accelerated upper-level north Pacific subtropical jet and a deepened and southeastward shifted Aleutian low, both leading to increased precipitation in the U.S. southwest. The latter on the Mediterranean basin drying is a result of future changes stemming from a present day weaker Atlantic Meridional Overturning Circulation (AMOC) that is energetically related to the double-ITCZ and models with weaker AMOC in the historical simulations tend to simulate a weaker AMOC response to warming. Given the double-ITCZ bias in E3SM, an analogy may be drawn between the implications of the double-ITCZ bias on the precipitation response to warming (i.e., difference between future and historical simulations) discussed by Dong et al. (2021) and the implications of the double-ITCZ bias on the precipitation bias in the historical simulations (i.e., difference between E3SM simulations and MERRA2), which is our focus. More specifically, we focus on if and how the aforementioned processes related to a double-ITCZ bias (stronger subtropical jet and weaker AMOC) influence AR biases in E3SM while also noting other large-scale biases.

In our study, the U.S. southwest and neighboring areas in E3SM feature positive AR biases during the NDJFM period (Fig. 1f). Although Dong et al. (2021) looked at future projections of large-scale circulation and precipitation changes, we find similarities to the above features in the large-scale circulation and precipitation biases which can explain the sources of some of the AR biases; the same processes arising from the double-ITCZ in future simulations can occur in present day simulations, although, the wet-get-wetter response for the future simulations likely enhances the effect. Over the central north Pacific, we find E3SM features a stronger, southward shifted north Pacific jet (Fig. 7e and 7g) and deepened geopotential heights during the winter compared to MERRA2 (Fig. 7c). These circulation biases are consistent with the double-ITCZ bias in E3SM through the aforementioned interconnected processes and contribute to enhanced moisture transport (Fig. 7a) and thus positive AR biases on the southern flank of the north Pacific storm track and landfalling regions (U.S. southwest). The coastal U.S. southwest also features an area of enhanced atmospheric moisture (not shown) which is likely a result of an underestimation of west coast, subtropical stratocumulus clouds (Golaz et al. (2019) Fig. 4c) leading to increased downward radiation and thus increased

evaporation and moisture. While most of the stronger positive moisture transport anomalies are directed towards the U.S. west coast, the central Pacific low geopotential height anomalies also support weaker enhanced transport to Alaska/Siberia - an area of positive AR frequency bias. While the jet bias in the North Pacific may be partly explained by the double-ITCZ bias in E3SM, biases in the subtropical jet are also noticeable in other regions that may or may not be related to the double-ITCZ. For example, over India, another area with positive AR frequency biases during NDJFM, the subtropical jet is similarly stronger and southward shifted compared to MERRA2. With the subtropical jet aimed more south of the Himalayas and the Tibetan plateau, an enhanced trough develops over Central Asia (Fig. 7c) which weakens the offshore winter monsoon and generates positive moisture transport anomalies onshore during the winter (Fig. 7a).

During MJJAS, the austral winter, the southern hemisphere subtropical jet is slightly stronger and shifted equatorward (Fig. 7f and 7h). This strengthening and equatorward displacement is not as strong nor as coherent as the northern hemisphere jet shift. The strengthening and/or shift is most apparent around 20° S which is just south of the positive AR frequency biases over Australia, the south Pacific, and southern Africa (Fig. 1i). Moisture transport anomalies (Fig. 7b) at these locations are poleward and westerly; they are supported by low geopotential height anomalies to their south (Fig. 7d). For the northern hemisphere MJJAS, the westerlies in general are enhanced in E3SM equatorward of about 40° N until the tropics. In contrast to the NDJFM response, the MJJAS upper-level winds are strengthened over the north Atlantic stretching east all the way to east Asia. Over the northwestern Pacific, there are positive AR frequencies along the western boundary of the Pacific basin - a region associated with East Asian summer rainband. Enhanced summertime westerlies across the Tibetan plateau have been linked to an intensified pre-Meiyu rainband resulting from increased meridional stationary eddy circulation and moisture convergence downstream of the Tibetan Plateau in east Asia (Chiang et al. (2019)). The E3SM anomalies show strengthened westerlies over the Tibetan Plateau along with increased moisture convergence east of the Tibetan Plateau and increased transport poleward at the location of the east Asian rainband. The enhanced transports reach up to Alaska, supported by the low geopotential height anomalies over Siberia and Alaska. Another region with positive AR anomalies is the Arabian Peninsula. Moisture flux anomalies over this region seem to be due, in part, to a weakened Somali Jet and redirected Indian monsoon moisture. The positive geopotential height anomaly in the Arabian Sea supports moisture transports towards the Arabian Peninsula.

Annually, regardless of the season (although stronger during MJJAS), Antarctica features positive biases just offshore. We find the E3SM southern hemisphere polar jet to exhibit more meridional movement than MERRA2 during MJJAS (Fig. 7f). This enhances the southwesterly moisture transports (Fig. 7b) towards Antarctica on the eastern side of the low anomalies. The same geopotential height anomalies that support the subtropical AR biases are also responsible for this variable jet movement. Additionally, Golaz et al. (2019) also reports fully-coupled historical simulation Southern Ocean net radiation to be higher and SSTs to be ∼2 degrees C higher than observations. We find higher atmospheric moisture in the southern hemisphere (up to 2 kg m$^{-2}$; not shown) from the subtropics to Antarctica given this warm SST bias during both seasons. Together, these biases may contribute to the higher AR frequencies near the Antarctic coast.

From examining the major AR biases and large-scale conditions in E3SM regionally and seasonally in the previous sections, we find these features to be most significant: i) the double-ITCZ bias, ii) a stronger and/or equatorward shifted subtropical jet during boreal and austral winter, and iii) stronger westerlies during the northern hemisphere summer. As previously mentioned,

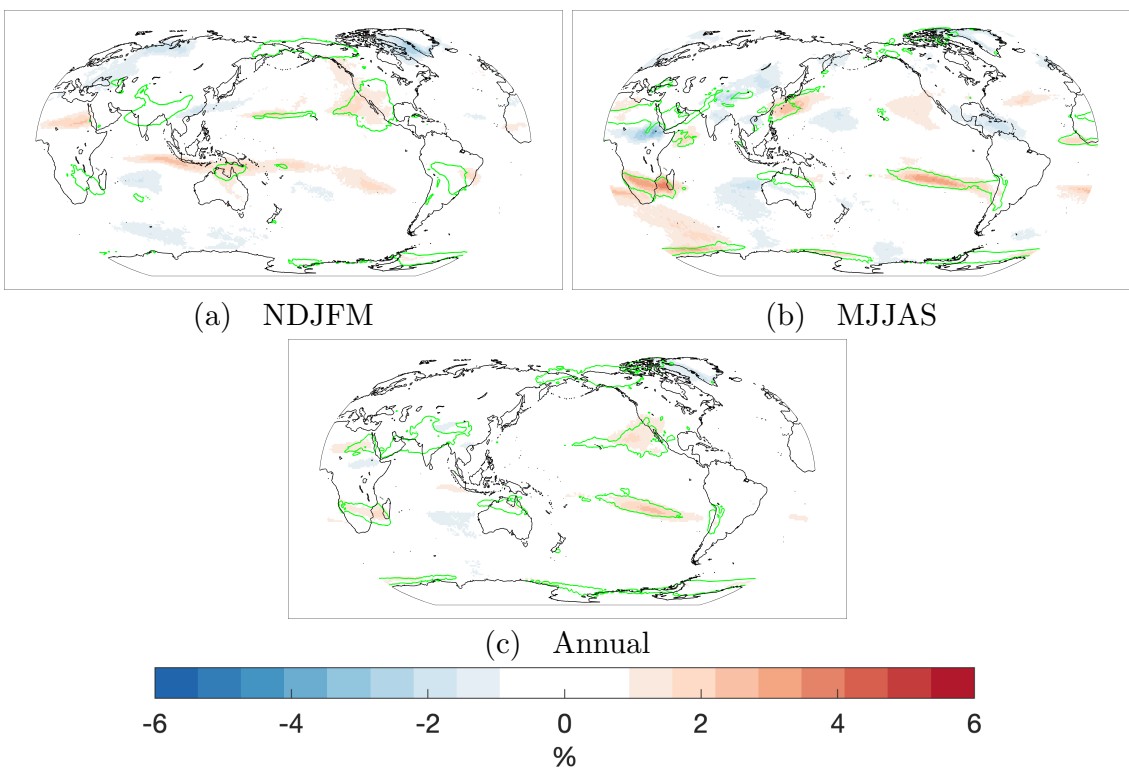

**Figure 8.** Historical ensemble (5 members) mean AR frequencies minus AMIP ensemble (3 members) mean AR frequencies. Contour lines for the seasonal and annual indicate the 2 % and 1.5 % positive AR frequency biases respectively from the corresponding biases in Fig. 1c, f, and i.

Dong et al. (2021) found that models with a double-ITCZ and associated wetting in the equatorial central Pacific to feature an
345 enhanced subtropical jet and weaker mean-state AMOC under global warming. The large-scale anomalies we have uncovered suggest that the double-ITCZ bias in E3SM may play a large role in some large-scale circulation biases such as the subtropical jet bias in North Pacific that contribute to the E3SM AR biases. We look for further evidence by isolating biases in the atmospheric model using the AMIP (Atmospheric Model Intercomposion Project) simulations and comparing them to the fully-coupled model for ARs. While atmospheric models may exhibit weak double-ITCZ biases, such biases are severely
350 exacerbated in fully-coupled models (Zhang et al. (2019)). This holds true for E3SM as can be seen in Golaz et al. (2019) Fig. 6b and 6c where the fully-coupled simulation has a far stronger double-ITCZ bias. Thus, by comparing AMIP and coupled simulations we can determine which biases are common to both simulations - implicating the EAM - or unique to the fully-coupled, historical simulation - implicating a coupling response (specifically the double-ITCZ response) or other components (e.g. the ocean component or sea ice).

355 We first compare the ensemble AR frequencies. The AMIP ensemble consists of 3 members (compared to the 5 members for the fully-coupled ensemble). For context, the AMIP frequencies have slightly better correlations and MAEs (as expected)

than the fully-coupled ensemble when compared to MERRA2 (e.g. annual ensemble frequency correlation is improved from 0.98 to 0.99 and annual MAE is improved from 0.60 % to 0.54 %). In Fig. 8, we subtract the AMIP ensemble frequencies from the fully-coupled ensemble frequencies. Common biases to both ensemble simulations - i.e. regions in Fig. 8 without fully-coupled minus AMIP biases (shading) but with fully-coupled minus MERRA2 biases (contours) - are the positive AR biases near elevated topography, India/Arabian Peninsula, central South America, southeastern Africa, and Alaska/Siberia. This suggests that these biases likely arise from the EAM. Golaz et al. (2019) reported that both AMIP and the fully-coupled historical simulations have excessive precipitation over elevated terrain as well as other precipitation biases.

We also identify several biases that are unique to the fully-coupled simulations. Compared to AMIP, the fully-coupled simulations have positive AR frequency biases (∼3 %) over the Pacific basin and over Africa at subtropical latitudes during the winter season of each hemisphere suggesting that the wintertime subtropical jet is affected going from AMIP to fully-coupled simulations - particularly on the equatorward flank. During NDJFM, the southwest region of the U.S. and the central north Pacific are zones of enhanced AR frequencies. This is an aforementioned area where the double-ITCZ bias response in models deliver excessive moisture (Dong et al. (2021)). For the southern hemisphere, positive AR frequencies are co-located with the double-ITCZ precipitation biases (see Fig. 5c). During MJJAS, the major biases are over the summer rainband region of east Asia, southern Africa, and the eastern subtropical Pacific in the southern hemisphere. Given the AR biases over the subtropics in the fully-coupled simulation compared to the AMIP simulation, we now examine the changes in the behavior of the subtropical jet between these two simulations.

In Fig. 9, we compare upper-level zonal winds (zonal wind at 200 hPa) globally for NDJFM and MJJAS. During NDJFM in the northern hemisphere, positive upper-level zonal wind anomalies in the fully-coupled simulation (Fig. 9a) generally match in location to the anomalies of E3SM compared to MERRA2 (Fig. 7e) while the negative anomalies, particularly over Asia, are weaker. This would suggest that coupling in E3SM is a major source of a stronger, slightly equatorward shifted, boreal winter subtropical jet. The zonal mean zonal winds (zonally averaged over the Pacific and Atlantic basin) show similar equatorward shifts when comparing the fully-coupled simulation to both MERRA2 and AMIP (Fig. 7g and Fig. 9c). The southern hemisphere differences are generally consistent with MERRA2 differences from the equator to the subtropics but the midlatitudes to the polar latitudes lack the southward shifted, enhanced westerlies over the southern ocean/Australia. The enhanced westerlies over this region are a bias common to both fully-coupled and AMIP simulations implicating the EAM.

The global MJJAS zonal wind (200 hPa) anomalies between the fully-coupled and AMIP simulations qualitatively matches the MJJAS anomalies when comparing to MERRA2 for most regions. A notable feature is the clear strengthening and south-ward shift of the subtropical jet seen over subtropical latitudes over much of the northern hemisphere - particularly from north Africa moving east to the Pacific basin. The enhanced westerlies over the Tibetan Plateau again correspond with increased AR activity downstream over the east Asian summer rainband region. West of and over southern Africa there is a band of enhanced westerlies in the fully-coupled simulation when comparing to both MERRA2 and AMIP (Fig. 7f and Fig. 9b). For both of these regions, the evidence points to the biases arising from coupling.

Lastly, we explore what physically causes the jet behavior shifts between fully-coupled and AMIP simulations. For NDJFM, the most significant change is a strengthening of the subtropical jet over the north Pacific and for MJJAS, the enhanced wester-

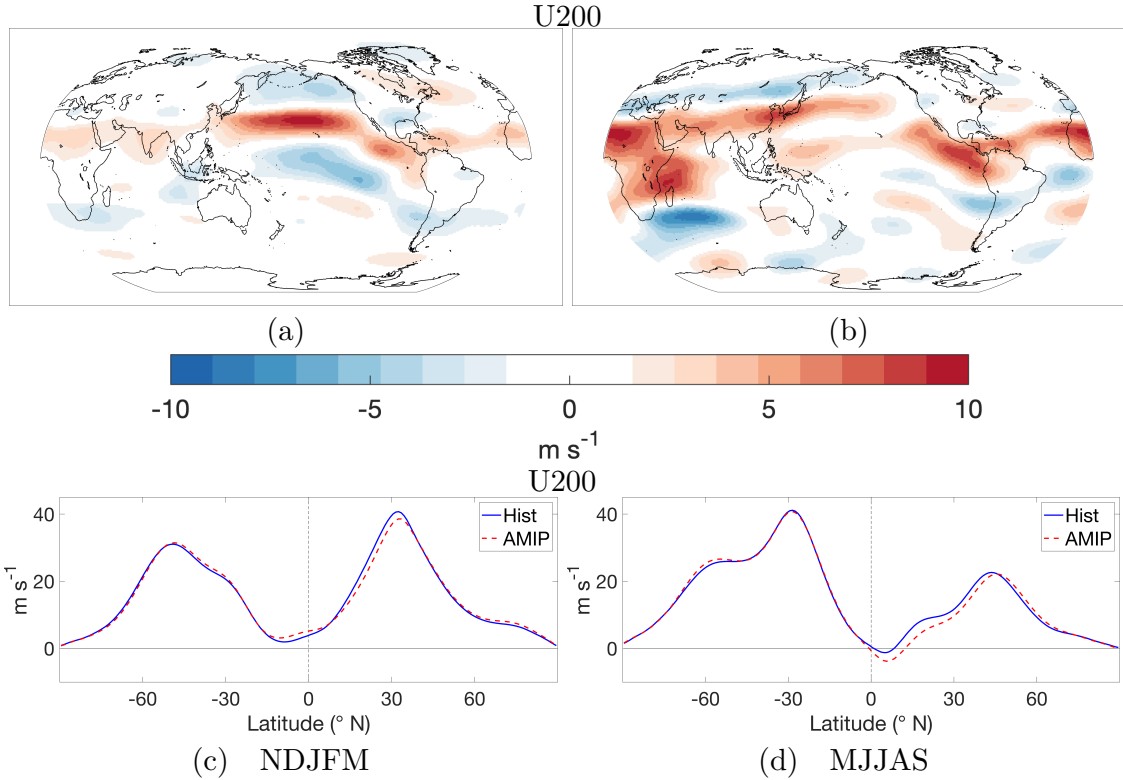

**Figure 9.** Seasonal differences between E3SM fully-coupled and AMIP: zonal wind at 200 hPa (top row) and zonal means over the Pacific and Atlantic basins (100-360° E) of zonal wind at 200 hPa (bottom row).

lies throughout much of the northern hemisphere. Building off the work of Dong et al. (2021), the two significant responses to a double-ITCZ bias that the authors uncovered in models are a strengthened subtropical Pacific jet in projections and a weaker mean-state AMOC in present day. We look for evidence that these responses occur for E3SM going from the AMIP to the fully-coupled simulation using upper-troposphere (500-200 hPa) temperatures differences (Fig. 10). Dong et al. (2021) found that due to the double-ITCZ, enhanced precipitation over the southern Central Pacific generated subtropical changes induced by latent heat release. This leads to enhanced upper-tropospheric warming over the subtropics (Dong et al. (2021) Fig. 2b), which increases the meridional temperature gradient locally, accelerating the subtropical jet along with a southeastward shift of the Aleutian low in projections. We find a similar upper-tropospheric response in temperature over the North Pacific (Fig. 10a); a striking patch of warming occurs in the same area of the subtropical north Pacific with a corresponding cool patch to the north. This enhances the north Pacific wintertime subtropical jet as seen in Fig. 7g.

Dong et al. (2021) also find evidence of weaker present-day AMOC in models with a double-ITCZ bias. While the authors focus on the implications for winter precipitation projections over the Mediterranean basin, in this study we examine whether a weaker mean state AMOC can enhance MJJAS westerlies. The MJJAS response is clearly different than that of the NDJFM

response. Fig. 10b reveals widespread cold anomalies throughout the northern hemisphere contrasted with some warm anoma-
lies throughout the southern hemisphere. The strongest cold anomalies are concentrated in a subtropical/midlatitude band over
north Africa stretching east to east Asia. The strong upper-troposphere cold anomalies at these latitudes increase the meridional
temperature gradient supporting an accelerated summertime subtropical jet. In fact, the band of cold anomalies sits just north
of the enhanced subtropical jet anomalies over Africa, Europe, and Asia (Fig. 9b). The hemispheric temperature contrast of
a warm southern hemisphere and a cold northern hemisphere is suggestive of a weaker AMOC (Liu et al. (2020)); a weaker
AMOC would deliver less cross-equatorial heat to the northern hemisphere causing the northern hemisphere to be cooler. The
weaker AMOC and the double-ITCZ are related as a double-ITCZ attempts to counteract less northward heat transport with
increased northward heat transport via a stronger southern hemisphere ITCZ (Zhang et al. (2019)).

We also examine the surface air temperature to verify a weaker AMOC in the fully-coupled simulation. Specifically, we look
for the classic AMOC "fingerprint" which consists primarily of a strong cold temperature anomaly over the subpolar Atlantic
Ocean and to a lesser degree as a warm temperature anomaly over the Gulf Stream (Caesar et al. (2018)). In Fig. A4, we
show surface temperature differences between the fully-coupled and AMIP simulations and find the AMOC "fingerprint" well
defined along with a hemispheric contrast in temperature. Liu et al. (2020) isolated the global surface air temperature response
to a weakened AMOC and find widespread northern hemisphere cooling and more modest southern hemisphere warming (Liu
et al. (2020) Fig. 2E). This is consistent with Golaz et al. (2019) and Hu et al. (2020), both of which reported a weaker AMOC
calculated directly from the ocean model output in the fully-coupled E3SM simulation when compared to observations.

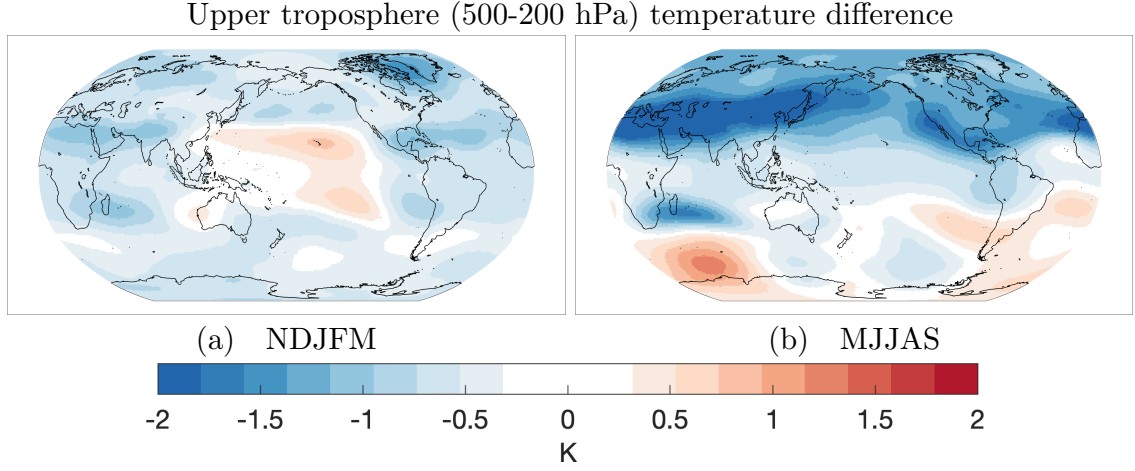

**Figure 10.** Seasonal differences between the fully-coupled E3SM and AMIP simulation for upper troposphere (500-200 hPa) temperature.

## 4   Conclusions

In this study, we have evaluated E3SM v1.0 at standard resolution for its ability to simulate ARs globally. We compared the
fully-coupled historical simulation to MERRA2 and began with an examination of global AR frequencies. We find that E3SM

is able simulate ARs with very high degrees of correlation and low MAEs annually (annual correlation 0.98; MAE 0.60 %) and seasonally (NDJFM correlation 0.98; MAE 0.72 %; MJJAS correlation 0.97; MAE 0.82 %). Amongst historical ensemble members, we determined that the internal variability of AR frequencies is low with the 5-member SD under 0.5 % for nearly all grid points. There are however, some biases (most notable when looking at relative differences) such as: i) positive biases occurring near the tropic/subtropic edge during both northern and southern hemisphere winters, ii) enhanced AR activity over

the Middle East, India, and the western boundary of the north Pacific basin during boreal summer, and iii) on the windward side of elevated terrain (e.g. Tibetan Plateau). AR characteristics are compared using probability distributions and we find the E3SM AR characteristics are generally consistent (shape and peak) with MERRA2. Some differences are that the median magnitude of mean AR IVT is higher and ARs tend to be slightly more zonal in E3SM. The generally higher IVT and stronger westerly jets in E3SM are likely the source of both characteristic differences.

E3SM distributions of AR precipitation show good agreement with MERRA2 although there is a clear bias resembling the double-ITCZ bias and excessive maritime continent precipitation. This manifests as reduced AR precipitation along the equatorial Pacific and excessive AR precipitation just off the equator in the double-ITCZ regions. AR precipitation fractions reveal a bias in E3SM to attribute excessive precipitation just off coast of the western U.S. and western Chile as well as over the region near northern Africa/the Middle East/India.

An examination of the large-scale conditions relevant to ARs in E3SM reveals these features to be most significant in producing AR biases in E3SM: i) the double-ITCZ bias, ii) a stronger and/or equatorward shifted subtropical jet during boreal and austral winter, and iii) enhanced westerlies during the northern hemisphere summer. The work of Dong et al. (2021) showed there is a significant relationship in models with a present-day double-ITCZ bias to have a stronger projected north Pacific subtropical jet as well as a weaker present day AMOC. Given the clear double-ITCZ bias in the fully-coupled E3SM, we

investigated whether the interconnected processes of the North Pacific jet, Aleutian low, and AMOC with a double-ITCZ were present in the E3SM historical simulation and if they could explain AR biases. Analysis of the E3SM large-scale circulation biases identified biases in the subtropical jet during both NDJFM and MJJAS that could contribute to the AR precipitation biases. The strengthened and slightly equatorward-shifted North Pacific jet and the impact on AR precipitation in the U.S. Southwest is consistent with the signature identified by Dong et al. (2021) as related to the double-ITCZ during winter.

Motivated by the analysis of large-scale circulation biases and their general correspondence with the AR precipitation bias, we further compared the fully-coupled (strong double-ITCZ bias) and AMIP simulations (no to weak double-ITCZ bias) to isolate the changes that occur when moving from an atmosphere-only model to a fully-coupled model while specifically looking for evidence of a stronger subtropical jet and weaker AMOC. The analysis suggests that the AR frequency biases over elevated terrain, India/Arabian Peninsula, central South America, and Alaska/Siberia can be attributed to the EAM, as similar biases are

found in both AMIP and coupled simulations. Biases arising from coupling or other model components include the positive AR frequencies over the north Pacific subtropics and U.S. southwest region during NDJFM and over the east Asia summer rainband region and the southern hemisphere eastern subtropical Pacific during MJJAS. These coupling biases suggest that the model responses to a double-ITCZ revealed in Dong et al. (2021) could be the source even in present-day simulations. We show evidence that the physical processes leading to a stronger north Pacific subtropical jet during NDJFM and enhanced northern

hemisphere westerlies during MJJAS are consistent with Dong et al. (2021). The north Pacific subtropical jet is enhanced via increased, upper-troposphere temperature gradients generated through teleconnections induced by enhanced heat release in the equatorial Pacific Ocean related to the double-ITCZ. Reducing the double-ITCZ bias remains an open issue in modeling, but previous studies suggests that improvements to parameterizations of boundary-layer turbulence and convective schemes can reduce this bias (e.g. Song and Zhang (2018); Lu et al. (2021)) which in turn could improve the water vapor transport and AR biases seen in E3SM as well as other GCMs.

On the other hand, the enhanced northern hemisphere westerlies during summer are due to a band of strong cold anomalies in the upper-troposphere stretching east from north Africa to east Asia. The upper-troposphere temperature differences reveal a hemispheric temperature contrast with a cool northern hemisphere and warm southern hemisphere bias - suggestive of a weaker AMOC. We find that the fully-coupled simulation does indeed have a weaker mean state AMOC evidenced by the AMOC "fingerprint" in surface temperature comparisons, which is consistent with the weak AMOC reported by Golaz et al. (2019) and Hu et al. (2020) based on analysis of the ocean circulation in the coupled simulations.

We note, however, that the cold bias in the northern hemisphere and the opposite bias in the southern hemisphere may also be contributed by the strong model response to aerosol forcings, as found by Golaz et al. (2019). Aerosol forcing is strongest over the northern hemisphere midlatitudes (Hansen et al. (1998); Ma et al. (2012); Friedman et al. (2013)) during spring through summer and during MJJAS is indeed when we see the strongest signal in cold anomalies. Another contributing factor for the interhemispheric temperature contrast could be from the delayed warming - related to E3SM's strong aerosol forcing - in the coupled historical simulation between 1960-1990 which keeps the global surface air temperature lower than observations until about 2010. The long period of delayed warming could reduce the interhemispheric temperature asymmetry signal from climate change which has amplified warming of the northern hemisphere (Friedman et al. (2013)). The cool northern and warm southern hemisphere bias in the coupled E3SM simulation may also explain why the northern MJJAS hemisphere jet strengthening and shift is more significant than the southern hemisphere as the upper-level temperature gradients are increased and decreased over the subtropical latitudes for the North and South hemispheres respectively. More generally, biases in the subtropical jet in E3SM may be contributed by other sources of model biases besides the double-ITCZ and related weak AMOC. Future analysis including the high resolution E3SM simulation (Caldwell et al. (2019)) may offer additional insights on AR and large-scale circulation biases, as AMOC is noticeably stronger at high resolution compared to the low resolution simulations analyzed here.

This study not only provides a comprehensive, global overview of AR representation in the fully-coupled historical E3SM v1.0 simulation that should give users of E3SM confidence in its ability to realistically simulate ARs but also seeks to understand how and why some biases are present. While we have framed this analysis through the lens of ARs, the biases in large-scale conditions are relevant to other phenomena and also provide potential areas of improvement in the EAM and the fully-coupled simulations. Evaluating AR frequency biases in other CMIP5/6 GCMs to reanalysis, such as those used in O'Brien et al. (2021), suggests that the AR biases associated with a double-ITCZ may be more general than found in E3SM alone. Analysis of the high-resolution simulation and future projections by E3SM is useful to further understand model biases and their implications for projecting future changes in AR frequency and intensity and extreme precipitation.

*Code and data availability.* E3SM and MERRA2 datasets used in this study are publically available at https://esgf-node.llnl.gov/projects/e3sm/ and https://disc.gsfc.nasa.gov/datasets?project=MERRA-2 respectively. The tARget v3 algorithm is available at https://doi.org/10.25346/S6/B89KXF.

## 5    Appendix

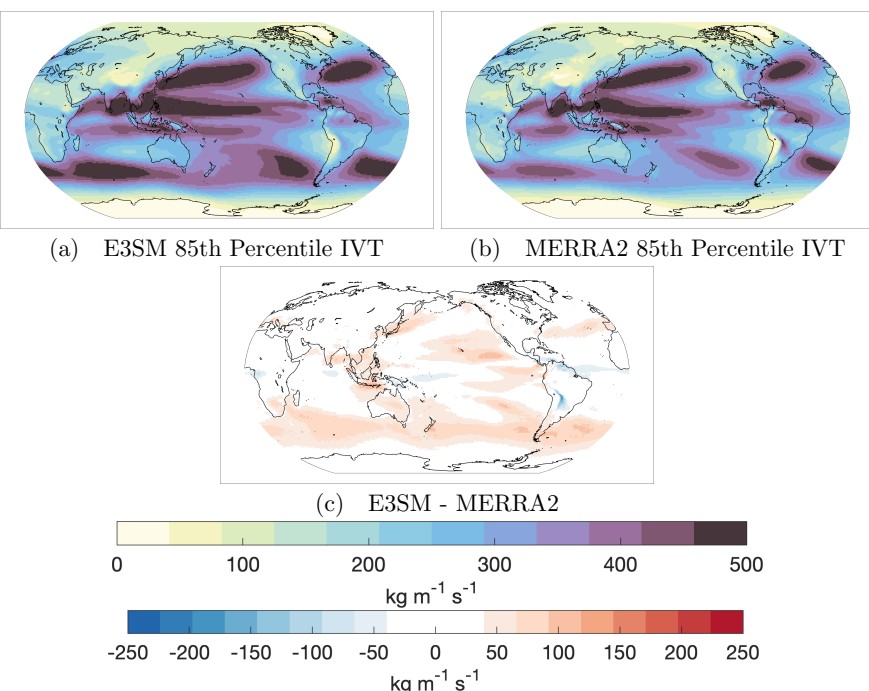

(a)   E3SM 85th Percentile IVT          (b)   MERRA2 85th Percentile IVT

(c)   E3SM - MERRA2

**Figure A1.** Annual mean 85th percentile IVT for (a) E3SM and (b) MERRA2. The difference (E3SM minus MERRA2) is shown in (c).

*Author contributions.* SK and LRL conceived the project idea and plan. SK, LRL, and JC contributed to the investigation and design of the methodology. BG provided the tARget v3 atmospheric river detection algorithm software and provided support to SK on applying it. LRL
and JC both supervised SK throughout the project. All authors contributed to the writing and reviewing of the manuscript. SK performed the formal analysis and created the visualizations.

*Competing interests.* The authors declare that they have no conflict of interest.

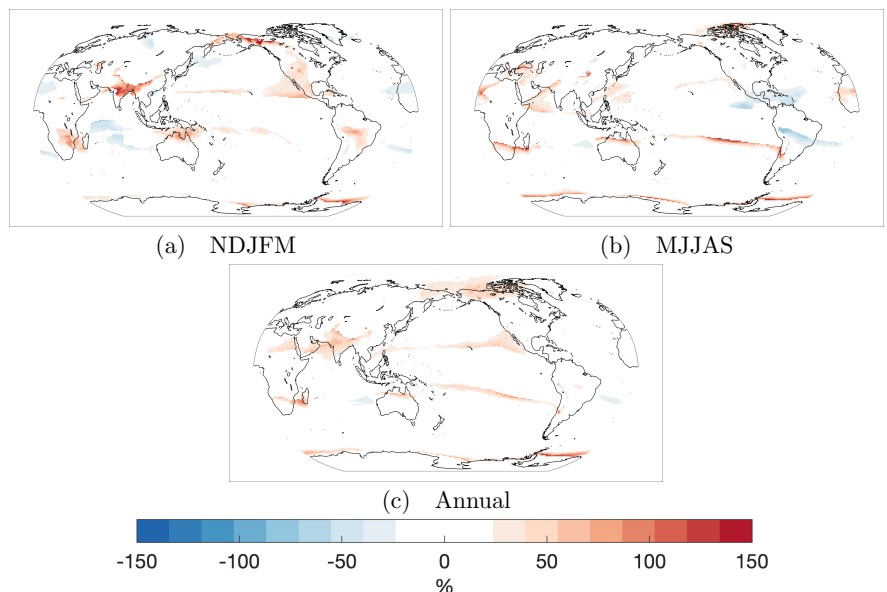

(a) NDJFM  (b) MJJAS

(c) Annual

**Figure A2.** AR relative frequency differences (as opposed to absolute frequency differences) by percentage corresponding with Fig. 1c, f, and i. Only gridpoints with MERRA2 AR frequencies (absolute) of at least 3% are shown (regions with very low AR frequencies can show relative differences of 100% due to a single extra timestep).

*Acknowledgements.* SK and LRL were supported by the Office of Science, U.S. Department of Energy Biological and Environmental Research as part of the Regional and Global Model Analysis program area. SK was also supported by the Geography Department at the University of California, Berkeley. We acknowledge National Energy Research Scientific Computing Center (NERSC), a U.S. Department of Energy Office of Science User Facility located at Lawrence Berkeley National Laboratory, operated under Contract No. DE-AC02-05CH11231, for the allocation of computational resources which enabled us to perform the data analysis. PNNL is operated for the Department of Energy by Battelle Memorial Institute under contract DE-AC05-76RL01830.

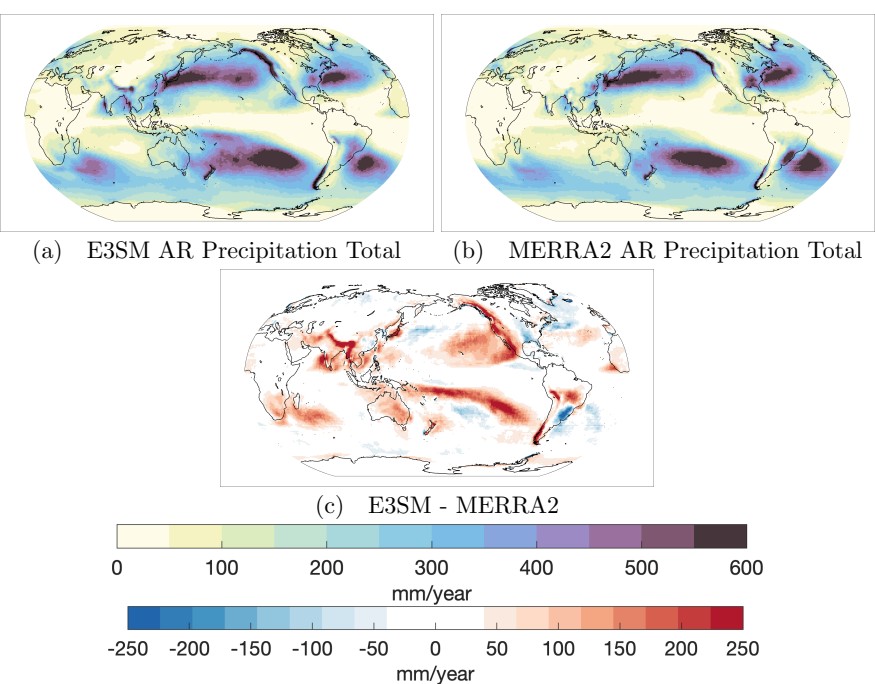

(a)  E3SM AR Precipitation Total        (b)  MERRA2 AR Precipitation Total

(c)  E3SM - MERRA2

**Figure A3.** Mean annual total AR precipitation for (a) E3SM and (b) MERRA2. The difference is shown in (c).

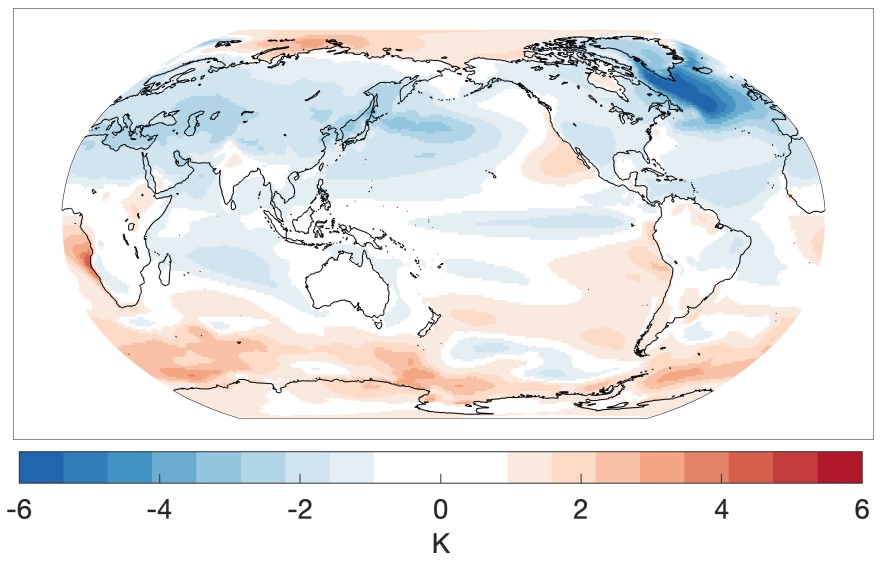

**Figure A4.** Annual surface temperature differences between the fully-coupled E3SM simulation and the AMIP E3SM simulation.

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
