# Peer review of "Atmospheric River Representation in the Energy Exascale Earth System Model (E3SM) Version 1.0"

_Geoscientific Model Development, 2021_

## Author Comment (AC3)

**Response to Reviewers #1: Atmospheric River Representation in the Energy Exascale Earth System Model (E3SM) Version 1.0**

Sol Kim, L. Ruby Leung, Bin Guan, and John C. H. Chiang

February 19, 2022

We thank the reviewers very much for their suggestions, comments, and time on our submitted manuscript. We have addressed all points raised by the reviewers (copied here and shown in bold text), and include our responses below. Any new or altered figures are included at the end.

**1 Reviewer 1**

**The authors evaluated the simulations of atmospheric rivers in the model E3SM and explored the relevant physical and dynamical processes. However, I have some major concerns as well as many other comments listed below. Before the author solve these concerns, this manuscript might not be ready to publish.**

**1.1 Major concerns**

1. **Lines 112-120: The authors compared the AR frequency between MERRA2 and E3SM. Although the authors clarified that "All % differences mentioned below ... are absolute differences, not relative difference", the description is misleading. For example, "there is a slight positive bias (1-3%) in the E3SM frequency near the edge between the tropics and subtropics ..." I do not think 1-3% (absolute difference in AR frequency) is a "slight positive bias". For example, while the AR frequency is below 10% over the southern California coastal region (Figs. 1a-b), the 1-3% bias (Fig. 1c) is large. In other words, the relative difference/bias of AR frequency over that region is larger than 10-30%. The bias in Chile is even larger (Fig. 1c). Please rewrite this part and the potential reasons for the large bias should be discussed.**

   We agree that relative differences are important to consider and we have added

a figure of relative differences ((E3SM ensemble frequencies - MERRA2 frequencies)/MERRA2 frequencies x 100) to the appendix but we only display the regions where MERRA2 AR frequencies are above 3%. The logic being that 3% frequency in the annual represents only 10-11 timesteps with an AR and even less during the seasonal (NDJFM and MJJAS). Areas with frequencies below 3% are generally confined to the highest latitude polar regions or very near the equator along with the Tibetan Plateau and Greenland. These low AR frequency regions can show relative differences of 100% due to a singe extra timestep. We also note that 1% of absolute AR frequency difference in the annual is 3-4 AR timestep differences.

With that said, there are important regions where ARs occur more frequently that must be considered including the suggested region off the coast of southern California and Mexico, over India/Himalayas, and generally at the subtropical/tropical edge. We have rewritten Section 3.1 (Results) to reflect these important relative difference regions and address their biases throughout Section 3.4.

2. **Line 122 and line 130: "... exhibits a close match ..." and "... in agreement with ..." Similar to my last comment, please be careful with these vague descriptions. For example, as the authors mentioned that in NDJFM over the west coast of North America, the AR frequency difference is 3-4% (line 125). That means the relative difference is roughly > 30-40% over the US West Coast (one of the most important area that affected by ARs) during the AR season (NDJFM). These model bias should be carefully examined and described.**

We also agree with the importance of this seasonal and regional bias (relative differences are 30-40% as commented) and we have rewritten Section 3.1 (Results) to emphasize the regions with large relative differences and made reference to Section 3.4 where we address the biases. To note, we focus on this specific bias throughout the rest of the paper in relation to the subtropical jet.

3. **Many statistical analyses (such as Figs 2 and 4) are conducted over the global domain. Those results are important. However, many useful information/signals might be smoothed out using the global domain. It would be helpful if the authors could conduct similar analyses for some regions with high frequency or high impacts of ARs, such as the west coast of North America in NDJFM. I believe many readers will be interested in the analysis for that kind of region (with large model bias as well as high social impacts), rather than a smoothed result for a global domain.**

While we are in agreement that a regional analysis of high impact areas would be interesting and worthwhile, we take the view of Reviewer #2 (Section 2.1, #4) that the global scope of analysis is more appropriate for an overview of AR representation in E3SM. To keep the scope of the paper more concise, we would like to keep the domain global and allow more targeted future studies to focus on the

regional biases based on the overview provided in this paper.

4. **I have some concerns and suggestions for Fig. 4. (a) In my understanding, in each panel the total probability of E3SM or MERRA2 should be 1. However, it seems like the total probability is much lower than 1. Please clarify. (b) In panel a, please put "x10$^4$" closer to the unit or the numbers in X-axis. (c) Please keep consistency in the number of decimals for the values of medians in each panel. (d) In caption, "sets of line" should be "sets of lines". (e) Please extend the Y-axis in panel e since it seems like the maximum value is higher than the probability of 8x10-3 (is the probability values at Y-axis correct?).**

Thank you for catching the error and for all the suggestions on Fig. 4. (a) Indeed the total probability should add to 1 for both E3SM and MERRA2. Placeholder NaNs in the data matrices were being included in the histogram counts causing the error which has been corrected. The probabilities now properly add to 1. (b) We have implemented your suggestion and added the x10$^4$ to the unit on the X-axis label. (c) Two decimal places are displayed for each of the median values to be consistent. (d) "sets of line" has been corrected to "sets of lines." (e) All the panel Y-axes have been extended going beyond the maximum value.

5. **Lines 227-233: "A large source of general E3SM precipitation biases come from ... suggesting certain large-scale circulation biases may have larger influence on AR frequency than the frequency of non-AR storms." These sentences are important to interpret the potential mechanism responsible for the model bias in AR precipitation. However, it is difficult to follow the logic. For example, how did the authors conclude that "... certain large-scale circulation biases may have larger influence on AR frequency than the frequency of non-AR storms" without analyzing the frequency and precipitation rate of non-AR storms?**

To clarify, we concluded that large-scale circulation biases may have a larger influence on AR frequency than non-AR storms by examining several things: general precipitation biases, AR precipitation biases, fractional precipitation attributed to ARs, and AR frequencies. Given that the two regions we mention in the text (off the coast of the U.S. southwest and around India/Pakistan) have positive AR precipitation biases, AR fractions, and AR frequencies but no general precipitation biases, suggested to us that circulation biases in these regions have a stronger effect on AR frequencies than non-AR precipitation producing storms. If the circulation biases affected non-AR storms in a similar manner, one could expect a bias in the general precipitation in these regions. However, the positive AR frequency biases (absolute and fractional preceipitation percentage) in these regions point to large-scale circulation influencing ARs more strongly. We have edited this section to be clearer on how we arrived at our conclusion.

6. **Fig. 5 shows the comparison of AR precipitation rate. How about the**

**total AR precipitation amount? For example, over the US West Coast E3SM has positive bias in both AR precipitation rate and AR frequency in NDJFM. I am curious how large the bias of total AR precipitation will be over there.**

Thank you for the suggestion - we have included a figure in the appendix of annual AR precipitation totals for E3SM and MERRA2 along with the difference between the two. As expected, areas with positive AR frequency biases and/or positive biases in fraction of precipitation attributed to ARs show positive annual AR precipitation total biases. The areas with the highest positive biases (up to 250 mm) are the west coast of North and South America, over the south Pacific where the double-ITCZ bias is, and near the Himalayas - all regions with positive AR frequency biases. To note, a single AR event making landfall on the west coast of North and South America can deliver well over 100 mm in a single day.

7. **I suggest the authors go through the manuscript to improve the writing. This manuscript would be easier to read if the authors could improve the writing. I listed some issues below, but there are more places that could be improved.**

We have gone through several rounds of editing specifically with the intention of improving writing and clarity. All sections of the paper have received revision and we have implemented all the comments suggested below.

**1.2 Other comments**

1. **Line 58: What is "standard resolution"?**

We have clarified that for E3SM, 'standard resolution' refers to 1° x 1° resolution.

2. **Lines 58-63: These two sentences provide a general background for the E3SMv1 performance, but they are vague. Please re-write and provide more details.**

Agreed - we have rewritten the lines as follows: The standard resolution model (1° x 1°) has been shown to credibly simulate earth's climate when evaluated by means of a standard set of Coupled Model Intercomparison Project Phase 6 (CMIP6) Diagnosis, Evaluation, and Characterization of Klima simulations which include a preindustrial control, historical simulations, and idealized $CO_2$ forcing simulations (Golaz *et al.* (2019)). A suite of atmospheric fields (e.g. net top-of-the-atmosphere radiation, surface air temperature, zonal winds, precipitation) in the historical E3SMv1 simulations were compared against observations to calculate root-mean-square-errors (RMSEs). When compared to an ensemble of 45 CMIP Phase 5 (CMIP5) models, E3SMv1's RMSEs were generally found to have lower errors than the median of the CMIP5 ensemble, and for many fields and seasons, in the lowest (best) quantile.

3. **Line 79: is "daily data" daily mean or instantaneous?**

"Daily data" is the daily mean data and we have updated the text to reflect this.

4. **Line 80: "Five ensemble members ...". Please clarify the difference between the ensemble members, as well as the motivation to use the five ensemble members.**

For the E3SM historical simulation, a total of five ensemble members are available from the CMIP6 archive. These historical simulations use the initial conditions from a 500 year preindustrial control run with the first historical run using January 1st of year 101. Subsequent historical ensemble members are branched every 50 years (Golaz *et al.* (2019)). We chose to use the five available members to both analyze the internal variability of E3SM related to ARs and to generate ensemble AR frequency means for comparison to reanalysis. We have updated the paper to reflect the above and also describe how the AMIP ensemble members are generated.

5. **Line 87: "AMIP", please spell out the full name when it is used for the first time.**

We have added "Atmospheric Model Intercomparison Project (AMIP)" to this line.

6. **Line 88: "CMIP6 DECK simulations", please define DECK.**

We have added "Diagnosis, Evaluation, and Characterization of Klima (DECK)" to the first reference of DECK in the introduction (Section 1).

7. **Lines 105-106: "This means the threshold is calculated separately for MERRA2 and the E3SM simulation." Is there any large difference in the IVT threshold between MERRA2 and E3SM? The difference in the IVT threshold (85th percentile) could be a part of the model bias.**

We have calculated the annual ensemble mean IVT threshold (85th percentile) for E3SM and compared it to the IVT threshold for MERRA2 and included a figure in the Appendix. E3SM's IVT threshold shows positive biases over the southern hemisphere westerly jet where there are known warmer SST biases (Golaz *et al.* (2019)) and positive vertically integrated water vapor (not shown) biases. There are additional IVT biases over the north Pacific and north Atlantic where the subtropical jet has a strong bias. We include discussion of these IVT threshold biases when examining the large-scale biases in Section 3.4.

8. **Line 110: "All % differences ..." should be "All percentage differences ..."**

Fixed to 'percentage.'

9. **Line 152: "SDs are consistent with MERRA2." This sentence is too vague.**

   Agreed - we have rewritten to: "E3SM SDs, for all ensemble members and for all periods, are within 0.18% of MERRA2."

10. **Line 161: "... AR frequencies are well ¡ 1.0 %." Difficult to understand.**

    Full sentence now reads: "Analysis of the coefficient of variation (not shown), calculated at each grid globally as the ratio between ensemble AR frequency SD and mean ensemble frequency, shows that the annual SDs are well below 10 % of the ensemble mean AR frequencies for virtually all grid points barring a few grid points over the equator and Antarctica where ARs are very rare (annual AR frequencies are <1.0 %)."

11. **Line 162: "The seasonal SDs reveal sources of the higher annual SDs." Higher than which SDs? I saw the annual SD is obviously lower than the NDJFM and MJJAS SDs in Fig. 3.**

    The intent was to communicate that the regions where SDs are higher in the annual figure have seasonal sources - not to imply that the annual SDs are higher than the seasonal SDs. To make the writing clearer, we have opted to remove this sentence.

12. **Lines 165-168: "MJJAS SDs (Fig. 3b peak for 1.5 % over various regions of ... " Difficult to follow. Please rewrite.**

    We missed closing the parenthesis to (Fig. 3b) and have clarified this section. It now reads: "MJJAS SDs (Fig. 3b) have maxima of ∼1.5 % over various regions of the Asia summer monsoon - the Arabian Sea, over the Philippines, and east Asia. This suggests that during MJJAS, differences in monsoon, MJO, or subtropical jet behavior may drive AR frequency variability between various E3SM ensemble members."

13. **Line 168: "In general, the northern hemisphere shows more internal variability." This is an interesting result, but do the authors have any idea about the potential reasons?**

    We have rewritten to: "In general, the northern hemisphere shows more internal variability; the regions of seasonal SD maxima implicate ensemble variability in the behavior of the north Pacific subtropical jet and the northern hemisphere monsoons as potential reasons."

14. **Line 170: "... using a single historical simulation ... " Why did the authors use a single historical simulation? How did the authors select the single simulation?**

Given that our ensemble analysis showed low internal variability amongst histori­cal members and a manual check of several ensemble member distributions showed virtually no differences in all characteristics, we decided to use a single historical run as it also eased processing and storage demands. We arbitrarily selected the first historical ensemble member.

15. **Line 171: "The distribution of all the ARs . . . " Do the authors mean the "characteristics"?**

   The sentence now reads: "The distributions for the various AR characteristics are shown in Fig. 4."

16. **Line 172-173: "All characteristics show strong similarities in shape and peak at the same values, barring magnitude of mean IVT (4e)." I do not understand this sentence, what are the "same values"? I do not understand the logic to mention 4e (Fig. 4e?) here either?**

   The 'same value' we refer to is the bin value on the histogram and Fig 4e was referenced because it is the only AR characteristic for E3SM that does not peak in probability at the same bin as MERRA2. We have clarified this sentence to read: "All characteristic show strong similarities in the distribution shape and peak in probability at the same bin values aside from magnitude of mean IVT (Fig. 4e)."

17. **Lines 175-177: These two sentences are difficult to follow. Please rewrite.**

   We have tried to make this section clearer and it now reads: "The length (Fig. 4a) and width (Fig. 4b) distributions of ARs in E3SM are generally consistent with MERRA2 and with previous characterizations of AR geometry (e.g. Guan and Waliser (2015)). The median length and width of ARs in E3SM are ∼3% longer (3501 km compared to 3397 km) and wider (658 km compared to 639 km) than in MERRA2. The E3SM AR length/width ratio distribution and median (Fig. 4c) are in good agreement with MERRA2."

18. **Lines 199-200: "For annual AR precipitation, E3SM reproduces the . . . " Please be careful, there might be large differences in the distributions and magnitudes of AR precipitation if using different precipitation data. "Reproduce" might be too vague.**

   We have edited the sentence to be more explicit about the data and it now read: "For annual AR precipitation, E3SM is able to simulate the general global distri­bution and magnitude of AR precipitation characterized in previous studies, which use data from the Global Precipitation Climatology Project version 1.2, (Guan and Waliser (2015) and Ralph *et al.* (2020)) as well as with MERRA2, but with some differences."

19. **Line 202, "The western coasts . . . produce higher rates of AR precipi­tation in E3SM." "Produce" might not be suitable here.**

We agree and changed "produce" to "have."

20. **Line 226-227: ". . . model bias in the subtropical jet would affect precipitation . . . , as both are influenced by the jet and storm tracks." Please provide reference.**

We have added the following references which relate the subtropical jet to both AR and non-AR precipitation: Shields and Kiehl (2016), Zhang and Villarini (2018), Wahl *et al.* (2019), Ralph *et al.* (2020).

21. **Lines 250-252: "Although Dong et al. (2021) looked at future projections of large-scale circulation and precipitation changes . . . which can explain the sources of some of the AR biases." I do not understand the logic to use the results of future projections from Dong et al. (2021) to explain "the sources of some of the AR biases" in the historical simulations in this study. Many factors and even mechanisms may change under climate change.**

We agree that this connection to Dong et al. (2021) can be improved in our writing. To clarify, Dong et al. (2021) used the present day double-ITCZ bias to find a mechanistic link to projected enhanced future precipitation estimates over the U.S. west coast. They found that the main mechanism by which the double-ITCZ bias can affect U.S. west coast precipitation is to enhance subtropical upper tropospheric warming (induced by increased latent heat release) which leads to an increased meridional temperature gradient and thus strengthens the subtropical Pacific jet and deepening the Aleutian low. We find this same mechanistic link arising from the double-ITCZ bias in E3SM. We have rewritten this paragraph along with others to make this connection more clear.

22. **Fig. 8: It would be helpful if the authors could add contours to show the distribution of AR frequency in the historical simulations.**

Thank you for the suggestion - this makes it far easier for readers to see the comparisons. Similar to Fig. 6c we have added contours to the figure to show E3SM minus MERRA2 frequency biases above 2.0% and 1.5% in the fully-coupled historical runs for the seasonal and annual respectively. We have edited the text to reflect this update.

**2   Reviewer 2**

**In this manuscript, the authors compare historical simulations of global AR frequency in an Earth System Model (E3SM) to AR frequency in a reanalysis dataset (MERRA2). The authors highlight differences in model depictions of global AR frequency and provide some physical insights into these biases.**

The manuscript is detailed, and I appreciate the authors' attempt to put the E3SM biases into context. However, I feel that the manuscript could benefit from several major changes prior to publication. I base my recommendation on the general comments listed below.

**2.1 General Comments**

1. **The methodology section is rather sparse on details, especially those related to E3SM, MERRA2, and the AR detection algorithm. For E3SM and reanalysis, which output fields are obtained? Is the daily data instantaneous (if so, at what time?) or daily-averaged? For the algorithm, a few additional details (e.g., regarding geometric criteria) would be helpful.**

   Following your suggestion, as well as Reviewer #1's, we have included more details on all three sections of the methodology. For E3SM, we further describe the performance of E3SM compared to other CMIP5 models, how the ensemble members are generated for both fully-coupled and AMIP simulations, what 'standard resolution' refers to, and which output fields are obtained. We also include that the daily data is a daily-average. For MERRA2, we included the output fields obtained along with a few more sentences providing background on MERRA2. We have added a more pertinent details regarding the detection algorithm including geometric and directional criteria.

2. **Regarding the choice of AR detection algorithm, recent intercomparison studies have shown that climate model simulations of AR activity vary based on choice of AR detection algorithm. While the authors are justified in choosing the Guan and Waliser algorithm, I think the authors should acknowledge possible uncertainty in results owing to the choice of algorithm.**

   We agree that we did not provide enough context regarding uncertainty related to detection algorithm. We have acknowledged this uncertainty in the methodology section (Section 2.3 'Atmospheric River Detection Algorithm') and in the conclusion section (Section 4). References included related to the uncertainty are: Lora *et al.* (2020), O'Brien *et al.* (2020), Rutz *et al.* (2019), Shields *et al.* (2019), and Shields *et al.* (2018).

3. **I agree with Referee #1 that the characterization of differences in AR frequency should be in a relative sense vs. an absolute sense. Given that ARs only occur for at most 15-20% of the time steps, these absolute differences are considerable. I agree that this section should be overhauled to reflect the relative differences.**

   Agreed with both reviewers - we have included a response to Referee #1 in Section

1.1 Major concerns #1. In short, we acknowledge the importance of relative differences as they can be large (>30% at times) and added a figure of relative differences to the paper along with a rewrite of Section 3.1 AR Frequency. We reference Section 3.4 where we address the biases of the areas of high relative differences.

4. **Regarding the choice of domain, while I agree with Referee #1 that a regional domain would be useful for more targeted studies, I think that the global domain is appropriate for the scope of this paper, which I interpret as a first step in understanding depictions of AR frequency in E3SM.**

   We agree with this view and thank you for providing this perspective. We have responded to Reviewer #1 in Section 1.1 Major concerns #3.

5. **While the authors provide a good initial overview of the biases seen in E3SM, the authors do not provide much insight into potential pathways toward improving the model. I am specifically thinking of the parameterization of convection, PBL turbulence, microphysics, etc., which could have an influence on the depiction of water vapor transport. At the very least, the authors should note that the model physics could play a role in depicting ARs and precipitation, even if this is left to future research.**

   We appreciate this point and believe adding some context on potential model improvements would strengthen the paper. Our main result suggests that the double-ITCZ bias and follow on effects lead to much of the AR biases in E3SM. While the double-ITCZ remains an open issue in modeling, previous studies, in line with your thoughts, suggests that improvements to parameterizations of boundary-layer turbulence and convective schemes can reduce the double-ITCZ bias (e.g. Song and Zhang (2018); Lu *et al.* (2021)) and this in turn could improve the water vapor transport/AR biases seen in E3SM. We include this discussion in Section 4 (Conclusions).

**2.2   Specific Comments**

1. **Lines 40-54: The authors may consider adding a recent article, O'Brien et al. (2021), which analyzes changes in AR counts and size in a future climate within the CMIP5/6 models:**

   **O'Brien, Travis Allen and Wehner, Michael F and Payne, Ashley E. and Shields, Christine A and Rutz, Jonathan J. and Leung, L. Ruby and Ralph, F. Martin and Marquardt Collow, Allison B. and Guan, Bin and Lora, Juan Manuel and et al., (2021) Increases in Future AR Count and Size: Overview of the ARTMIP Tier 2 CMIP5/6 Experiment. JGR A. https://agupubs.onlinelibrary.wiley.com/doi/10.1029/2021JD036013**

Thank you for this suggestion and reference. We have included the reference in the suggested paragraph along with accompanying edits.

2. **Lines 78-95: Which fields are analyzed for the model and reanalysis datasets? At the bare minimum, the specific humidity and wind fields would be required to calculate IVT. Are there additional fields downloaded?**

   We have added the fields obtained from both E3SM and MERRA2 in Section 2.1 and 2.2 respectively of the Data and Methods. For E3SM the fields are: total (vertically integrated) zonal/meridional water flux, total (convective and large-scale) precipitation rate (liquid + ice), total (vertically integrated) precipitatable water, zonal wind at 200 hPa, and geopotential height at 500 hPa. We included corresponding fields along with the appropriate long name for MERRA2. We did not calculate IVT ourselves using wind and humidity but rather used the model derived IVT provided as an output field by both MERRA2 and E3SM.

3. **Lines 96-106: Given the importance of the choice of AR detection algorithm to this study, I suggest that the authors provide some additional details about the detection algorithm (even though the details have already been published elsewhere).**

   Agreed - we have added further details about the detection algorithm and also acknowledged the uncertainty associated with the choice of AR detection algorithm (as per your suggestion in 2.1 General Comments #2).

4. **Lines 109-110: How are the ensemble mean AR frequencies calculated? Are ARs detected for each ensemble member and the frequencies averaged? Or are the ARs detected using the ensemble-average IVT?**

   The former - ARs are detected separately for each ensemble member and the frequencies are averaged for the ensemble. We have clarified this in the lines suggested.

5. **Figure 1: While the absolute differences are important, relative differences are likely more interesting given the relative infrequency of ARs, even over the midlatitude storm track.**

   We include a figure of the relative differences (annually and seasonally) and have rewritten Section 3.1 (AR Frequency) to reflect this. As suggested, there are significant relative differences, particularly on the equatorward side, of the storm tracks.

6. **Figure 2: This is a very interesting plot, but visually, the data points are difficult to discern. There is a lot of "empty" space in these diagrams. Would it be possible to either a) make the data points larger or b) only show a fraction of each diagram? Zooming in somehow would be very helpful for the reader to help discern the slight differences in correlations**

**between ensemble members.**

We had a similar thought but unfortunately, suggestion a) does not improve clarity since larger data points just increase the overlap between the points and b) is not possible with the software we used. A question asking if it would be possible to zoom in on the correlation coefficient received this answer from the creator of the MATLAB toolbox used to generate the Taylor Diagram.
https://github.com/PeterRochford/SkillMetricsToolbox/wiki/FAQ:

"Q14. Is it possible to limit the correlation range within a Taylor diagram when there is a cloud of data with similar correlation values so as to make the plot clarity better?

A14. It is not possible to limit the correlation range because the Taylor diagram is based on the similarity of the equation relating the various statistics with the Law of Cosines."

7. **Lines 165-168: Do features like the Indian Monsoon, large-scale tropical convection, or TCs get detected as ARs?**

While AR activity and monsoon activity in the Asian monsoon region both peak during the summer months and are related to each other due to their ability to transport significant amounts of water vapor, they are distinct features (Liang and Yong (2021)). The AR detection criteria employed by Guan and Waliser (2019) distinguishes ARs from strong monsoonal flows due to both the 85th percentile IVT requirements (intense monsoonal flow IVTs peak at 350 kg/m/s while the 85th percentile IVT during summer months is greater than 500 kg/m/s in the Asian monsoon region) and the geometric constraints (e.g. length greater than 2000 km with length to width ratio greater than 2) (Liang and Yong (2021)). Tropical convection, as seen in Fig. 1, is not mistaken for atmospheric rivers using this detection method as ARs are extremely infrequent, if occurring at all, in the tropical regions. TCs are also not detected as ARs using this detection method. In Section 2.2 (AR Identification) of Guan and Waliser (2019) and in their Supporting Information Fig. S3, TCs are filtered out as they will generate a ring-shaped axis. The algorithm is developed to then initiate a search and deletion of the circular portion of the IVT object based on finding the pixels where IVT goes cyclonically around a common center. This method allows for the identification of ARs associated with TCs but will not include the TC as part of the detected AR object. We include some of these details in Section 2.3 where we describe the detection algorithm.

8. **Lines 171-173: Are the feature-averaged values (namely, feature-averaged IVT) weighted by latitude?**

There is no weighting by latitude for the feature-averaged IVT. However, the IVT threshold used by the detection algorithm (Guan and Waliser (2019)) uses location- and season-dependent 85th percentile IVT.

9. **Line 178: Is the feature's centroid based on area alone, or is there a weighting based on IVT intensity?**

   For the AR detection algorithm (Guan and Waliser (2015); Guan and Waliser (2019)), the centroid is IVT- and area-weighted. It would be the center of mass if IVT were mass density.

10. **Lines 184-185: I don't agree with this statement. While this may be the case here, tropical ARs could have weaker moisture transport due to a lack of strong winds aloft. I suggest the authors clarify this statement.**

    We agree this statement needs to be reworked and thank you for suggesting the clarification. While the annual 85th percentile IVT values equatorward of the subtropics (30N and 30S), are in general stronger than those found in the subtropics the same is true for the IVT values poleward of the subtropics in the midlatitudes. Thus, the fact that there are more ARs close to the tropics and less ARs in the midlatitudes does not explain the stronger mean IVT values in E3SM. Examining the IVT biases in E3SM however, shows that there are positive IVT biases in general for E3SM compared to MERRA2 with some of the strongest biases near the tropics. We have rewritten this statement to also acknowledge the IVT biases and as stated in previous responses, have added a figure of the annual IVT 85th percentile of E3SM, MERRA2, and the difference of those two in the paper appendix.

11. **Figure 7: Although the details are provided in the figure caption, I suggest adding labels to these panels for quicker/easier interpretation.**

    We have added text indicated which field is being shown in the figure rows.

12. **Lines 322-323: Are these individual AMIP and fully-coupled simulations or an ensemble average of each?**

    These are based on the ensemble average (5 fully-coupled and 3 AMIP simulations).

13. **Figures 8-10: As with Figure 7, it would be helpful if the panels were labeled. Also, is AMIP subtracted from fully-coupled, or vice-versa?**

    We agree this would be helpful for readers and have added labels of the fields visualized for Fig. 7, 9, and 10 but have opted to keep Fig. 8 labeled with just the seasons since Fig. 8 is just frequencies and to keep consistent with previous frequency figures (Fig. 1). AMIP is subtracted from the fully-coupled and we state this in both the Fig. 8 caption and include this in the text for clarity.

**2.3    Technical Corrections**

1. **Line 51: "but only a few"**

   Corrected the line.

2. **Lines 54, 60, 78: Parentheses around citation**

   Added parentheses around the citations at the given lines as well as any other missing ones.

3. **Line 87: Spell out the AMIP abbreviation**

   Corrected to: "Atmospheric Model Intercomparison Project (AMIP)"

4. **Line 179: Use "fewer" rather than "less" here**

   Agreed - we have made the change.

5. **Line 321: Is this "southern Africa", as opposed to the country?**

   Yes - we have changed it to southern Africa.

6. **Line 344: Use "occur" rather than "occurs" here**

   Corrected

**References**

Golaz, J.-C., Caldwell, P. M., Van Roekel, L. P., Petersen, M. R., Tang, Q., Wolfe, J. D., Abeshu, G., Anantharaj, V., Asay-Davis, X. S., Bader, D. C., *et al.* (2019). The doe e3sm coupled model version 1: Overview and evaluation at standard resolution. *Journal of Advances in Modeling Earth Systems*, **11**(7), 2089–2129.

Guan, B. and Waliser, D. E. (2015). Detection of atmospheric rivers: Evaluation and application of an algorithm for global studies. *Journal of Geophysical Research: Atmospheres*, **120**(24), 12514–12535.

Guan, B. and Waliser, D. E. (2019). Tracking atmospheric rivers globally: Spatial distributions and temporal evolution of life cycle characteristics. *Journal of Geophysical Research: Atmospheres*.

Liang, J. and Yong, Y. (2021). Climatology of atmospheric rivers in the asian monsoon region. *International Journal of Climatology*, **41**, E801–E818.

Lora, J. M., Shields, C., and Rutz, J. (2020). Consensus and disagreement in atmospheric river detection: Artmip global catalogues. *Geophysical Research Letters*, **47**(20), e2020GL089302.

Lu, Y., Wu, T., Li, Y., and Yang, B. (2021). Mitigation of the double itcz syndrome in bcc-csm2-mr through improving parameterizations of boundary-layer turbulence and shallow convection. *Geoscientific Model Development*, **14**(8), 5183–5204.

O'Brien, T. A., Payne, A. E., Shields, C. A., Rutz, J., Brands, S., Castellano, C., Chen, J., Cleveland, W., DeFlorio, M. J., Goldenson, N., *et al.* (2020). Detection uncertainty matters for understanding atmospheric rivers. *Bulletin of the American Meteorological Society*, **101**(6), E790–E796.

Ralph, F. M., Dettinger, M. D., Rutz, J. J., and Waliser, D. E. (2020). *Atmospheric Rivers*, volume 1. Springer.

Rutz, J. J., Shields, C. A., Lora, J. M., Payne, A. E., Guan, B., Ullrich, P., O'brien, T., Leung, L. R., Ralph, F. M., Wehner, M., *et al.* (2019). The atmospheric river tracking method intercomparison project (artmip): quantifying uncertainties in atmospheric river climatology. *Journal of Geophysical Research: Atmospheres*, **124**(24), 13777–13802.

Shields, C. A. and Kiehl, J. T. (2016). Atmospheric river landfall-latitude changes in future climate simulations. *Geophysical Research Letters*, **43**(16), 8775–8782.

Shields, C. A., Rutz, J. J., Leung, L.-Y., Ralph, F. M., Wehner, M., Kawzenuk, B., Lora, J. M., McClenny, E., Osborne, T., Payne, A. E., Ullrich, P., Gershunov, A., Goldenson, N., Guan, B., Qian, Y., Ramos, A. M., Sarangi, C., Sellars, S., Gorodetskaya, I., Kashinath, K., Kurlin, V., Mahoney, K., Muszynski, G., Pierce, R., Subramanian, A. C., Tome, R., Waliser, D., Walton, D., Wick, G., Wilson, A., Lavers, D., Prabhat, Collow, A., Krishnan, H., Magnusdottir, G., and Nguyen, P. (2018). Atmospheric river tracking method intercomparison project (artmip): Project goals and experimental design. *Geoscientific Model Development Discussions*, **2018**, 1–55.

Shields, C. A., Rutz, J. J., Leung, L. R., Ralph, F. M., Wehner, M., O'Brien, T., and Pierce, R. (2019). Defining uncertainties through comparison of atmospheric river tracking methods. *Bulletin of the American Meteorological Society*, **100**(2), ES93–ES96.

Song, X. and Zhang, G. J. (2018). The roles of convection parameterization in the formation of double itcz syndrome in the ncar cesm: I. atmospheric processes. *Journal of Advances in Modeling Earth Systems*, **10**(3), 842–866.

Wahl, E. R., Zorita, E., Trouet, V., and Taylor, A. H. (2019). Jet stream dynamics, hydroclimate, and fire in california from 1600 ce to present. *Proceedings of the National Academy of Sciences*, **116**(12), 5393–5398.

Zhang, W. and Villarini, G. (2018). Uncovering the role of the east asian jet stream and heterogeneities in atmospheric rivers affecting the western united states. *Proceedings of the National Academy of Sciences*, **115**(5), 891–896.

[Figure]

Fig.04

[Figure]

Fig.07

[Figure]

(a)   NDJFM                                    (b)   MJJAS

(c)   Annual

Fig.08

[Figure]

Fig.09

Upper troposphere (500-200 hPa) temperature difference

(a)  NDJFM  (b)  MJJAS

K

Fig.10

[Figure]

(a)  E3SM 85th Percentile IVT  (b)  MERRA2 85th Percentile IVT

(c)  E3SM - MERRA2

kg m$^{-1}$ s$^{-1}$

kg m$^{-1}$ s$^{-1}$

Fig.A1

[Figure]

(a)  NDJFM  (b)  MJJAS

(c)  Annual

Fig.A2

[Figure]

(a)  E3SM AR Precipitation Total  (b)  MERRA2 AR Precipitation Total

(c)  E3SM - MERRA2

Fig.A3

---

## Referee Report (RR1)

Review of Revisions to "Atmospheric River Representation in the Energy Exascale Earth System Model (E3SM) Version 1.0" by Kim et al.

**General Comments:**

I appreciate the author's consideration of my major concerns as well as my additional suggestions. I feel that the authors have sufficiently bolstered the methodology section, including their description of the E3SM model and the Guan and Waliser AR detection algorithm. The authors have also addressed the concerns of both reviewers related to the use of absolute vs. relative differences in comparing E3SM to MERRA2. The authors now place the differences into proper context and, in doing so, give the reader a much more realistic perspective on the biases in E3SM. Finally, though finding pathways toward improving these biases in E3SM is important, I acknowledge that proposing and evaluating specific model improvements is beyond the scope of this study. However, I appreciate the authors' reference to possible model improvements in the conclusion.

Overall, I have no additional major concerns and feel that this manuscript is now suitable for publication.